



# Sensitivity of young water fractions to hydro-climatic forcing and landscape properties across 22 Swiss catchments

**Jana von Freyberg[1,2], Scott T. Allen[1], Stefan Seeger[3], Markus Weiler[3] and James W. Kirchner[1,2]**

[1] Department of Environmental Systems Science, ETH Zurich, Zurich, Switzerland

[2] Swiss Federal Institute for Forest, Snow and Landscape Research (WSL), Birmensdorf, Switzerland

[3] Faculty of Environment and Natural Resources, University of Freiburg, Freiburg i. Br., Germany

*Correspondence to*: jana.vonfreyberg@usys.ethz.ch

**Abstract**

The young water fraction $F_{yw}$, defined as the proportion of catchment outflow younger than ca. 2-3 months, can be estimated directly from the amplitudes of seasonal cycles of stable water isotopes in precipitation and streamflow. Thus, $F_{yw}$ may be a useful metric in catchment inter-comparison studies that investigate landscape and hydro-climatic controls on streamflow generation. Here, we explore how $F_{yw}$ varies with catchment characteristics and climatic forcing, using an extensive isotope data set from

22 small- to medium-sized (0.7 - 352 km²) Swiss catchments. We find that flow-weighting the tracer concentrations in streamwater resulted in roughly 26 % larger young water fractions compared to the corresponding un-weighted values, reflecting the fact that young water fractions tend to be larger when catchments are wet and discharge is correspondingly higher. However, flow-weighted and un-weighted young water fractions are strongly correlated with each other among the catchments. They also

correlate with terrain, soil and land use parameters, as well as with measures of hydrologic response. Within individual catchments, young water fractions increase with discharge, indicating an increase in the proportional contribution of faster flowpaths at higher flows. We present a new method to estimate the linear slope of this relationship, which we call the discharge sensitivity of $F_{yw}$. Among the 22 catchments, discharge sensitivities of $F_{yw}$ are highly variable and only weakly correlated with $F_{yw}$ itself,

implying that these two measures reflect catchment behaviour differently. Based on strong correlations between the discharge sensitivity of $F_{yw}$ and several catchment characteristics, we suggest that low discharge sensitivities imply greater persistence in the proportions of fast and slow runoff flowpaths as catchment wetness changes. High discharge sensitivities, on the other hand, imply the activation of different dominant flowpaths during precipitation events, such as when subsurface water tables rise into

more permeable layers and/or the river network expands further into the landscape.





## 1    Introduction

Naturally occurring variations in stable water isotopes ($\delta^{18}O$, $\delta^2H$) or chemically passive solutes (e.g., chloride) are commonly used in catchment studies to track the flow of water and to gain insight into catchment storage and mixing behaviour (Buttle, 1994; Kendall and McDonnell, 1998; Klaus and

McDonnell, 2013).  Many catchment studies use these tracers to estimate time-averaged travel time distributions (TTD's), to characterize the heterogeneity of flow pathways, and to estimate mobile catchment storage (e.g., Benettin et al., 2015; Birkel et al., 2011; Hrachowitz et al., 2009; Staudinger et al., 2017).  TTD's are usually inferred from concentrations of conservative tracers, such as stable water isotopes in precipitation and streamwater using lumped-parameter models (McGuire and McDonnell,

2006).  Because the mean transit time expresses the ratio between catchment storage and the average flow rate, it is widely used in catchment inter-comparison studies (e.g., Hrachowitz et al., 2009; McGuire et al., 2005).  However, estimates of mean transit time can be biased and unreliable, especially for spatially heterogeneous catchments (Kirchner, 2016b; Seeger and Weiler, 2014).  Instead, the young water fraction $F_{yw}$ – i.e., the fraction of water that is younger than a specified threshold age – has

recently been proposed as a more reliable measure of water age in heterogeneous catchments (Kirchner, 2016a, b).  Young water fractions with a threshold age of roughly 2-3 months can be obtained directly from the amplitudes of the seasonal cycles of the stable water isotopes in precipitation and streamwater.

The amplitudes of the seasonal isotopic cycles in precipitation and streamwater can be estimated directly from the isotope measurements themselves, or by volume-weighting those measurements by the

corresponding precipitation or discharge rates.  Precipitation isotopes should generally be volume-weighted to prevent small precipitation events, potentially with anomalous isotope values, from substantially influencing the calculated seasonal precipitation isotope cycle.  Higher streamflows should typically correspond to larger young water fractions, for the simple reason that flow peaks typically follow intense rainfall and contain more recent precipitation than base flows (e.g., Kirchner, 2016b; von

Freyberg et al., 2017).  Hence, the flow-weighted average young water fraction (here denoted $F_{yw}^*$) is expected to be higher than the unweighted young water fraction ($F_{yw}$).  Both $F_{yw}$ and $F_{yw}^*$ are calculated over periods of a year or longer, and represent the average catchment behavior over that time.  In calculating the unweighted $F_{yw}$, each unit of time counts equally, and benchmark tests using a nonstationary lumped catchment model confirm that the calculated $F_{yw}$ should accurately reflect the

time-averaged fraction of young water in discharge (Kirchner, 2016b).  By contrast, in calculating the flow-weighted $F_{yw}^*$, each unit of flow counts equally, and benchmark tests confirm that the calculated $F_{yw}^*$ reflects the cumulative volume of young water, as a fraction of the cumulative volume of discharge, over the corresponding period (Kirchner, 2016b).  Although $F_{yw}^*$ and $F_{yw}$ have previously been

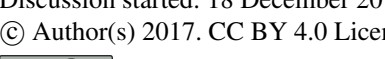



compared in benchmark tests, a systematic evaluation based on tracer data from natural catchments has not yet been done.

Another analytical decision that affects the interpretation of $F_{yw}^*$ and $F_{yw}$ relates to whether snowpack storage is considered to be part of catchment storage, or not. If one measures precipitation to the snow

surface as the catchment input, then snowpack accumulation and melt are implicitly included in catchment storage (e.g., Staudinger et al., 2017). In this case, comparisons of seasonal cycles in precipitation and streamflow should reflect the young water fraction resulting from the combination of snowpack and subsurface storage. Alternatively, if one uses precipitation and snowmelt arriving at the soil surface as the catchment input (for example, with melt pan lysimeters, or modeled snowpack

outflows), then snowpack accumulation and melt are implicitly excluded from catchment storage. In this case, comparisons of seasonal cycles in streamflow and sub-snowpack catchment input should reflect the young water fraction resulting from subsurface storage alone. Because the total catchment storage in the first case (including snowpack storage) is larger than the subsurface storage alone, the resulting young water fractions are expected to be smaller. Previous studies that estimated young water

fractions in snow-dominated watersheds (Jasechko et al., 2016; Song et al., 2017) did not differentiate between these two concepts of catchment storage and simply used incoming precipitation as one end-member in the young water fraction calculations, thus implicitly considering snowpack storage as part of catchment storage (as in the first case outlined above). This approach is practical in view of the challenges of measuring or modeling snowmelt and its isotopic composition. However, it is still

unclear whether, in cases where snowmelt can be modeled or measured, explicitly considering snowmelt as a catchment input would significantly alter young water fraction estimates.

Isotopic inputs to specific catchments can be estimated from nearby long-term monitoring stations using various spatial interpolation methods (e.g., Jasechko et al., 2016; Seeger and Weiler, 2014). These interpolation methods differ in their assumptions about temperature- and elevation-dependent isotope

fractionation effects, and their treatment of seasonal snowpack storage. Based on a global database of $\delta^{18}O$ in precipitation, Jasechko et al. (2016) calculated the seasonal cycle coefficients and their standard errors for each station and interpolated them to generate a global grid of the seasonal amplitude coefficients. These interpolated coefficients were volume-weighted by the spatial pattern of precipitation over each catchment. To generate a high-resolution precipitation isotope map for

Switzerland, Seeger and Weiler (2014) interpolated $\delta^{18}O$ in monthly precipitation from long-term monitoring stations in central Europe, using an elevation-gradient approach. They combined their interpolation method with an energy-balance-based snow model to estimate the liquid input to the soil surface at monthly temporal resolution. An alternative approach, which is presented in the Supplement builds from the Jasechko et al. (2016) method with an additional step that accounts for the residuals of





the observations from the fitted seasonal cycles. This method does not account for snow accumulation and melt. Both interpolation methods have been successfully validated with real-world isotope measurements, and thus may be particularly useful for estimating young water fractions in catchments where no long-term precipitation isotope time series exist.

Because the young water fraction can be estimated from sparse and irregular tracer data, it has been suggested as a useful metric for catchment inter-comparison studies (Kirchner, 2016a). Young water fractions were used in a global analysis of 254 watersheds, revealing large spatial variability in young streamflow, which correlated inversely with average topographic gradients and water table depths (Jasechko et al., 2016). Jasechko et al. hypothesized that steeper landscapes are associated with more

pervasive rock fracturing, deeper infiltration, and reduced shallow lateral flow, all of which would reduce the young water fraction in steep terrain. However, the correlation between topographic steepness and young water fractions was highly scattered, indicating that other factors are also involved. Jasechko et al. (2016)'s catchments were mostly larger than 1000 km$^2$ (25$^{th}$ percentile 1753 km$^2$, median 10,800 km$^2$) and thus were probably affected by a complex interplay of landscape characteristics,

climatic variability and human impacts. Identifying landscape and climatic drivers that potentially control catchment storage behaviour may be easier in small- to medium-sized catchments with near-natural streamflow regimes (Holko et al. (2015)).

In the present study, we use seasonal cycles in δ$^{18}$O to estimate young water fractions for 22 small- to medium-sized catchments (0.7 - 352 km$^2$) in Switzerland. Because these catchments cover a wide

range of landscape and hydro-climatic conditions, we also evaluate how the young water fraction estimates are affected by i) the spatial interpolation method for precipitation isotopes, ii) the conceptual representation of snow storage, and iii) flow-weighting the streamwater isotope data. We test for correlations between the young water fraction and a wide range of landscape and hydro-climatic indices. We present a method for estimating the linear dependence of the young water fraction on the

streamflow regime, and propose that the slope of this relationship may be a diagnostic indicator of runoff generation processes.

## 2   Theoretical background: Young water fractions from seasonal cycles of stable water isotopes in precipitation and streamwater

The isotopic composition of precipitation follows a seasonal cycle (Feng et al., 2009). The damping

and phase shift of this seasonal cycle as it is transmitted through catchments (Figure 1) can be used to infer time scales of catchment storage and transport.

Sine-wave fitting can quantify the amplitude ratio $A_S/A_P$ and phase shift $\varphi_S$-$\varphi_P$ between the seasonal isotope cycles in precipitation and streamflow (the indices $P$ and $S$ refer to precipitation and





streamwater, respectively). The seasonal isotope cycles in precipitation and streamwater can be described by:

$$c_P(t) = A_P \sin(2\pi f t - \varphi_P) + k_P \quad \text{and} \tag{1}$$

$$c_S(t) = A_S \sin(2\pi f t - \varphi_S) + k_S \quad . \tag{2}$$

In Eqs. (1) and (2), $A$ is the amplitude (‰), $\varphi$ is the phase of the seasonal cycle (rad, with $2\pi$ rad equalling 1 year), $t$ is the time (decimal years), $f$ is the frequency (years$^{-1}$) and $k$ (‰) is a constant describing the vertical offset of the isotope signal.

If one assumes that the transit times of water through the catchment follow a particular transit time distribution, the mean transit time can be calculated as a function of the amplitude ratio $A_S/A_P$.

However, mean transit times inferred from seasonal tracer cycles in runoff from heterogeneous catchments are potentially subject to severe aggregation bias (Kirchner, 2016a). Alternatively, the amplitude ratio $A_S/A_P$ can be used to estimate the fraction of water younger than a specified threshold age. Compared to the mean transit time, this "young water fraction" is markedly less vulnerable to aggregation bias, and less sensitive to the assumed shape of the catchment transit time distribution

(Kirchner, 2016a, b). For a wide range of transit time distributions, the young water threshold age is approximately 2.3±0.8 months (Kirchner, 2016a).

Here, we use multiple linear regression to obtain the coefficients $a$ and $b$ in

$$c_P(t) = a_P \cos(2\pi f t) + b_P \sin(2\pi f t) + k_P \tag{3}$$

and

$$c_S(t) = a_S \cos(2\pi f t) + b_S \sin(2\pi f t) + k_S \quad . \tag{4}$$

The amplitudes $A_S$ and $A_P$ can then be determined by

$$A_P = \sqrt{a_P^2 + b_P^2} \quad \text{and} \quad A_s = \sqrt{a_s^2 + b_s^2} \quad . \tag{5}$$

We estimate the coefficients $a_S$, $b_S$, $a_P$, and $b_P$ by fitting Eqs. (3) and (4) using iteratively reweighted least squares (IRLS), a robust estimation method that minimizes the influence of any potential outliers

(an $R$ script with our IRLS code is provided with the Supplement). We volume-weight Eq. (3) to avoid giving undue weight to low-precipitation periods. To calculate the flow-weighted young water fraction ($F_{yw}^*$), we also weight Eq. (4) by stream discharge. Following Kirchner (2016a), we calculate young water fractions as the amplitude ratio $A_S/A_P$. Uncertainties in the calculated unweighted and flow-weighted young water fractions are expressed as standard errors ($SE$) and are estimated using Gaussian

error propagation.





### 3    Data set

The 22 study catchments cover areas between 0.7 to 351.2 km$^2$ and have mean elevations between 472
and 2369 m a.s.l (Table 1). Most of the sites are located in the Swiss Plateau and in the northern Alps,
where the geology is characterized by sedimentary rocks (limestones, sandstones, marls, marly shales,

conglomerates, breccias) and unconsolidated sediments (clay, silts, sands). In the southern Alps, two
high-elevation catchments (Dischmabach and Riale di Calneggia) are predominantly underlain by
metamorphic rock (mica shist, gneiss), and Ova da Cluozza is the only study catchment underlain by
dolomite rock (Figure 2a and b).

Land use at lower elevations (400‑800 m) is predominantly agriculture, while grassland and forests can

be found at elevations up to around 1400 m. Much of the area above 1700 m is characterized by
grasses, shrubs, and sparse vegetation. At the highest-elevation sites (Ova da Cluozza and
Dischmabach), up to ~2 % of the drainage area is covered by glaciers. At all sites, the human influence
on river discharge is small, resulting in near-natural streamflow regimes.

Switzerland is characterized by a humid to temperate continental climate with the Alps creating

climatically distinct subregions. The wettest regions can be found in the northern pre-Alps and Alps, as
well as in the Canton of Ticino south of the Alps. The driest regions are located in inner Alpine valleys
in the Cantons of Valais and Grisons (Figure 2c). Average annual precipitation rates for the 22
catchments range from 887 to 1853 mm based on observations from 2000 to 2015. To differentiate
between the hydro-climatic regimes of the catchments, we applied the three classes (snow dominated,

rainfall dominated and hybrid) proposed in Staudinger et al. (2017) (Table 1). Precipitation is
distributed more-or-less evenly throughout the year, although peak inputs to the soil surface (melt and
precipitation) are shifted towards spring and summer in all snow-dominated and some hybrid sites.

#### 3.1    Streamwater isotope data

Streamwater grab samples were collected approximately fortnightly at 21 sites between mid-2010 and

mid-2011 or longer (see Table 1 for exact dates). Oxygen isotope ratios ($\delta^{18}$O) were measured with a
Picarro isotope analyser (Picarro Inc., Santa Clara, CA, USA) at the University of Freiburg i. Br.,
Germany, and are reported here as $\delta$ values relative to the VSMOW standard. The measurement
accuracy for $\delta^{18}$O is 0.16‰. For the Rietholzbach catchment, fortnightly streamwater $\delta^{18}$O data were
provided by the Institute for Atmospheric and Climate Science at ETH Zurich.

#### 3.2    Hydro-climatic data

Daily discharge data for 18 of the 22 sites were provided by the Swiss Federal Office for the
Environment. Discharge measurements for the Aabach catchment were made available by the Amt für
Wasser, Energie und Luft (AWEL) of Canton Zurich. Discharge data for the Erlenbach, Vogelbach and





Lümpenenbach catchments were provided by the Swiss Federal Institute for Forest, Snow and Landscape research (WSL), Birmensdorf, Switzerland.

Meteorological data for each site and each 100m elevation band were interpolated from measurements taken by the national meteorological service of Switzerland (MeteoSwiss), using the model PREVAH

(Viviroli et al., 2009). Mean precipitation for each 100m elevation band was aggregated to obtain the area-weighted catchment average values.

### 3.3   Catchment properties

The average hydro-climatic properties at the sites were described by various indices, such as mean monthly values of discharge $\bar{Q}$ and precipitation $\bar{P}$, as well as mean daily precipitation intensity

$\bar{P}_{intensity}$. To quantify the variability of the flow regimes, we determined the average coefficient of variation of daily discharge ($CV_Q$). Average daily baseflow ($Q_{bf}$), which was used to obtain the quickflow index ($QFI=(Q-Q_{bf})/Q_{tot}$) from daily discharge ($Q$), was calculated with the "BaseflowSeparation" function in the *EcoHydRology* package (version 0.4.12) in *R*; we used a recursive digital filter parameter of 0.925 as recommended by Nathan and McMahon (1990). All of these hydro-

climatic indices were calculated for the duration of the streamwater isotope sampling campaigns, which varied between approximately 1 and 5 years (Table 1).

The seasonal variability of monthly precipitation for the years 2000‒2015 was expressed through the amplitude and the phase shift of a fitted sinusoidal function (Berghuijs et al., 2014):

$$P(t) = \bar{P}\big[1 + A_{precip} \sin\big(2\pi(t - \varphi_{precip})/\tau\big)\big] \tag{6}$$

where $P$ is the precipitation volume (mm/month), $\bar{P}$ is the average of $P$ (mm/month), $A_{precip}$ is the seasonal amplitude of precipitation (-), $t$ is the time (months), $\tau$ is the duration of a full seasonal cycle (12 months) and $\varphi_{precip}$ is the phase (months). The phase describes the offset from the beginning of the seasonal cycle, which is defined here as January 1st. The parameters $A_{precip}$ and $\varphi_{precip}$ were obtained from monthly precipitation data using Newton's method. Strong seasonality would be expressed in a

high amplitude value.

The hydro-climatic indices are to some extent redundant with one another. Unsurprisingly, mean monthly discharge ($\bar{Q}$) and mean monthly precipitation ($\bar{P}$) were significantly correlated with each other across the 22 sites. Furthermore, $\bar{Q}$ was significantly correlated with the seasonality of precipitation ($A_{precip}$), and the quick-flow index ($QFI$) was significantly correlated with the coefficient of

variation of daily discharge ($CV_Q$) (Table 4).

To quantify the geomorphological characteristics of the catchments, we used terrain indices (median flow path length $L$, median flow gradient $G$, the ratio $L/G$ and median Topographic Wetness Index $TWI$)



which were calculated previously by Seeger and Weiler (2014) for all 22 study sites using a digital elevation model with 25 m spatial resolution. In addition, we calculated the drainage density *DD* (the total channel length divided by the catchment area) based on the official river network from the topographical landscape model of Switzerland (swissTLM3D, ©2017 swisstopo; resolution 8 m or better).

Hydrologic soil properties were extracted from geospatial data provided by the Swiss Federal Statistical Office. This data set uses six soil properties – soil depth (Gründigkeit), large particle fraction (Skelettgehalt), water storage capacity (Wasserspeichervermögen), nutrient storage capacity (Nährstoffspeichervermögen), permeability (Wasserdurchlässigkeit) and soil wetness index (Vernässung) – to generate a map of 144 different soil classes. Each soil property is ranked from 0 (very low) to 5 or 6 (very high). For our analysis, we calculated the areal fractions of aggregated soil properties that are usually associated with fast runoff processes, i.e., low storage capacity (ranks 1‑3), low permeability (rank 1‑3), and high soil wetness index (i.e., saturated soils, ranks 4‑5).

The hydrogeological properties of the sites were obtained from the official geotechnical map of Switzerland (1:200000, ©2017 swisstopo). We extracted the areal fractions of low, intermediate and high groundwater productivity for each catchment. Representative groundwater table depths could not be determined for all sites due to their complex small-scale topographic and geologic heterogeneity. The hydrologic soil properties, as well as the hydrogeological properties of the individual sites are provided in the Supplement (Table S1).

Correlations between the catchments' young water fractions, hydro-climatic conditions and landscape properties were assessed with the Spearman rank correlation coefficient $\rho$ (Spearman, 1987). Following conventional practice, we consider correlations with $p<0.05$ to be statistically significant.

## 4 Methodological evaluation of the young water fraction framework

### 4.1 Comparing two methods for spatial interpolation of $\delta^{18}$O in precipitation

Values of $\delta^{18}$O in precipitation were not measured directly at the 22 study catchments. Instead, monthly $\delta^{18}$O values were interpolated from long-term observations at nearby monitoring sites (the Swiss network for Observations of Isotopes in the Water Cycle (NAQUA-ISOT), the Global Network of Isotopes in Precipitation (GNIP), and the Austrian Network of Isotopes in Precipitation (ANIP)). We used two different interpolation approaches (method 1 after Seeger and Weiler (2014), and method 2 as presented in the Supplement) and compared the resulting seasonal cycles of precipitation isotopes and their effects on the calculated young water fractions.





In the interpolation method of Seeger and Weiler (2014), a kriging interpolation of the available precipitation isotope values from 26 long-term monitoring sites was adjusted for local differences in elevation, using the monthly average elevation gradient of $\delta^{18}O$ in precipitation, estimated from three nearby isotope monitoring stations that cover a similar elevation range as the 22 study catchments.

With method 2, we fitted isotope data from 19 long-term monitoring stations to sine curves using least squares. We then constructed a multiple linear regression model to explain the best-fit sine parameters as functions of latitude, longitude, and elevation. These spatially varying sine parameters were used to construct interpolated seasonal cycle maps for all of Switzerland. These seasonal cycles were then adjusted using kriged interpolations of the monthly residuals of station measurements from their fitted

seasonal patterns, to account for non-sinusoidal isotope dynamics. For both interpolation methods 1 and 2, monthly isotope values were mass-weighted based on the monthly elevation-dependent precipitation volumes obtained from the PREVAH model (Viviroli et al., 2009). Snow accumulation and melt were not accounted for. Detailed descriptions of both interpolation methods can be found in Seeger and Weiler (2014) and in the Supplement.

Figure 3a shows that the seasonal precipitation isotope cycle amplitudes ($A_P$) obtained with both methods are similar for most catchments; the differences range from -1.34±0.21 ‰ (±$SE$, Mentue) to 1.35±0.29 ‰ (Dischmabach). The interpolation method 2 results in larger $A_P$ values for the sites Alp, Biber, Mentue, Sense and Ria di Caneggia, compared to the results of method 1 (Figure 3a). On the other hand, smaller $A_P$ values are obtained with the interpolation method 2 for three high-elevation sites,

Allenbach, Dischmabach, and Ova de Cluozza. Overall, $A_P$ spanned a range of 1.77 ‰ with the interpolation method 2, compared to a larger range of 3.47 ‰ with the method 1. Nevertheless, for most sites the differences in $A_P$ between the two methods are small compared to the absolute values of $A_P$, and thus the choice of the interpolation method only marginally affects the estimated young water fractions $F_{yw}$. For all sites, the absolute differences between the values of $F_{yw}$ calculated with the two

interpolation methods are below 0.06 and statistically insignificant (Figure 3b).

A systematic test of both interpolation methods using on-site, long-term precipitation isotope measurements would go beyond the scope of this study. However, Seeger and Weiler (2014) have successfully validated their interpolation model with isotope measurements from six stations, while we validated method 2 as described in the Supplement. Results from the two methods are likely to differ

because they make different assumptions about the changes in precipitation isotopic composition with elevation. For our objectives, however, it is convenient that these two different approaches yield different $A_P$ in several cases, because it allows us to show that this level of variability in $A_P$ has only minor effects on the calculated young water fractions. Our comparison thus demonstrates that both approaches for spatially interpolating $\delta^{18}O$ in precipitation yield consistent young water fraction

estimates for the 22 study catchments.



### 4.2 The effect of snow storage on the seasonal cycle amplitudes and phases of precipitation isotopes

At high-elevation sites with a seasonally cold climate, precipitation (and its isotopic signature) will be stored temporarily in the snow pack in winter, and will be released during the melt season. Thus,

significant volumes of depleted snow meltwater may reach the river system during spring and early summer, when the isotope signal of incoming precipitation is more enriched. As a result, the seasonal isotopic variation in water reaching the soil surface (rainwater and snowmelt) is likely to be smaller than the seasonal pattern in precipitation alone.

The spatial interpolation method of Seeger and Weiler (2014) was extended using an energy-balance-

based model to explicitly simulate the storage of winter precipitation in the snowpack, so that the input to the soil surface, and its isotope signal, can be described by a mixture of rainwater and snow melt (the "delayed input" scenario in Figure 4). The energy-balance-based model uses PREVAH simulations of air temperature, wind speed, incoming shortwave radiation and precipitation amount to predict the melt water amounts and their average isotopic compositions for each 100m elevation band (without

considering isotopic fractionation of the snowpack and snowmelt). Alternatively, the interpolation can also be carried out by ignoring snowpack as a separate storage, such that the catchment input is taken directly from the incoming precipitation and its isotopic composition (the "direct input" scenario in Figure 4). Figure 4a shows, as an example, the time series of input water flux and $\delta^{18}$O (not volume-weighted) at the Dischmabach catchment for both scenarios. The delayed release of depleted winter

precipitation from the snowpack ("delayed input" scenario) results in a smaller seasonal amplitude of the input tracer signal. However, when this input tracer signal is volume-weighted, the fitted seasonal amplitude ($A_P$) values are statistically indistinguishable between the "direct" and "delayed" input scenarios for 21 of the 22 sites (Figure 4b). This result arises because the "delayed" input scenario gives very little weight to winter inputs in snow-dominated catchments (because snowmelt volumes

during winter conditions are small), allowing the fitted cycles to deviate from the winter isotope values. The difference in $A_P$ for both scenarios is statistically significant only at the Schaechen catchment, which contains the highest-elevation snowpacks in our data set (elevation up to 3260 m a.s.l., Table S1). As a consequence, snowmelt at the Schaechen site is isotopically more depleted compared to the other, lower-elevation sites. For the hybrid and rain-dominated sites, the $A_P$ values are almost

indistinguishable between the two scenarios, either because snowmelt occurs early in the season when rainwater and snowmelt have similar isotopic signatures (i.e., hybrid catchments), or because the contribution of snowmelt is small compared to that of rainfall (rain-dominated catchments). The young water fractions $F_{yw}$ are virtually identical between the "direct input" and "delayed input" scenarios (Figure 4c).





As can be seen in Figure 4a, the delayed meltwater input shifts the seasonal isotope pattern toward later in the season. Thus the "delayed input" scenario results in later cycle phases ($\varphi_P$) compared to the "direct input" scenario (Figure 4d), with statistically significant differences for the five high-elevation, snow-dominated sites and for four hybrid catchments, Erlenbach, Lümpenbach, Sitter, and Vogelbach.

However, the "delayed input" scenario had a statistically significant effect on the phase shift between input and output ($\varphi_S$-$\varphi_P$) only at Dischmabach ($\varphi_S$-$\varphi_P$=0.7 months) and Ria di Calneggia ($\varphi_S$-$\varphi_P$=0.8 months) (Figure 4e), and in all cases the percentage change in the phase shift was small. In the analysis presented below, we use precipitation isotope values that explicitly account for snowpack accumulation and melt (i.e., the "delayed input" scenario) in order to be consistent with previous studies

where this data set has been used (Seeger and Weiler, 2014; Staudinger et al., 2017).

### 4.3 Comparing unweighted and flow-weighted young water fractions

We use the isotope and discharge data sets of the 22 catchments to estimate young water fractions from the ratios of the seasonal cycle amplitudes $A_S$ and $A_P$, with and without discharge-weighting ($F_{yw}^*$ and $F_{yw}$, respectively). Flow-weighting of the streamwater isotope values results in a roughly 25 % increase

in the fitted seasonal streamwater isotope cycle amplitudes $A_S$, relative to the un-weighted $A_S$ values for the same sites (Figure 5a). Statistically significant differences between unweighted and flow-weighted values of $A_S$ were found for Dischmabach, Emme, Mentue, Rietholzbach, and Sense, as well as the Alp, Erlenbach, Lümpenenbach, Vogelbach and Biber (which are all located nearby one another, and share similar characteristics). Flow-weighting the discharge isotope values yields young water fractions ($F_{yw}^*$)

that are around 26 % larger than those calculated from unweighted discharge values ($F_{yw}$) (Figure 5b, Table 3), because high flows generally contain more young water than base flows. Thus, the flow-weighted $F_{yw}^*$ range from 0.07±0.01 to 0.49±0.03 (±$SE$), whereas the un-weighted $F_{yw}$ range from 0.06±0.01 to 0.37±0.03. The average values of $F_{yw}^*$ and $F_{yw}$ were 0.22±0.02 and 0.17±0.02, respectively, meaning that approximately 1/5 of total discharge was younger than roughly 2.3±0.8

months (assuming gamma distributions with shape factors $\alpha$ ranging from 0.3 to 2). These results are within the range of young water fractions reported for rivers in mountainous regions in North America and central Europe by Jasechko et al. (2016).

Perhaps unsurprisingly, the effect of flow-weighting on $A_S$ (and young water fractions) is largest in catchments with highly variable flow regimes. Thus, the effect of flow-weighting is strongest at sites

with relatively large values of $CV_Q$ (coefficient of variation of daily discharge) and $QFI$ (quick-flow index; Nathan and McMahon, 1990). In such catchments, robust estimation of the flow-weighted $F_{yw}^*$ may require a smart sampling strategy that captures a representatively wide range of hydrologic conditions.




## 5 Relationships between young water fractions, hydro-climatic conditions and landscape characteristics

Our 22 study catchments are good candidates for an inter-comparison study, because they span wide ranges of topographic and hydro-climatic characteristics, while human impacts on their streamflow

regimes are limited. By correlating the catchments' young water fractions with their landscape and hydro-climatic characteristics, we aim to identify dominant controls on their hydrological behaviour and storage dynamics. Below, we present our results using flow-weighted young water fractions ($F_{yw}^*$); however, very similar results were also obtained for the unweighted values $F_{yw}$ (Table 4).

Young water fractions exhibit statistically significant positive correlations with the hydro-climatic

indices $\bar{Q}$, $\bar{P}$, $\bar{P}_{intensity}$, $CV_Q$ and $QFI$ (mean monthly discharge, mean monthly precipitation, mean daily precipitation intensity, coefficient of variation in daily discharge, quickflow index, respectively; Table 4, Figure 6). These correlations suggest that young water fractions tend to be highest in humid catchments where prompt runoff response is facilitated by fast flowpaths and/or high-intensity precipitation events. $F_{yw}^*$ was also significantly correlated with high values of drainage density ($DD$)

and low values of flow path length ($L$) (Table 4). There was also a significant negative correlation with the ratio of the flow path length to gradient ($L/G$), but as there is nearly zero correlation with $G$ itself, the correlation with $L/G$ apparently arises through $L$ alone. Drainage density is inversely proportional to median flow path length, so the strong positive correlation of $F_{yw}^*$ with $DD$ and negative correlation with $L$ can be viewed as two sides of the same coin. All else equal, high values of $DD$, and thus small

values of $L$, facilitate faster runoff, which is directly linked to higher values of $CV_Q$ and $QFI$.

A statistically significant inverse correlation ($\rho=-0.36$, $p<0.0001$) between $F_{yw}$ and the logarithm of the topographic gradient was found by Jasechko et al. (2016) for 254 sites across Europe and North America, with the surprising implication that steeper catchments have less (not more) young streamflow. Between our individual catchments, however, we find no correlation between $F_{yw}$ and

topographic gradient. This may be partly explained by the lack of low-gradient catchments among our study sites; our gradients span a range of 0.02-0.64 compared to ~0.0007-0.11 in Jasechko et al. (2016), and the correlation that they observe appears to be largely driven by sites with gradients less than roughly 0.01. Nevertheless, our data set fits within the global pattern found by Jasechko et al. (2016), and the median $F_{yw}$ of our 22 mostly high-gradient study catchments (0.16, 95 % confidence interval

0.10 – 0.21) is smaller than the global median (0.21, 95 % confidence interval 0.19-0.24) consistent with the gradient-dependence hypothesized by Jasechko et al. (2016). Other studies have identified catchment area as a major control on mean transit times (e.g., Dewalle et al., 1997; Soulsby et al., 2000), however, the negative correlation of $F_{yw}^*$ and $F_{yw}$ with catchment area only becomes significant ($\rho=-0.49$, $p<0.05$) when the five high-elevation, snow-dominated sites are omitted from the analysis





(Figure 6). The young water fractions of the remaining 17 sites were also strongly correlated with mean catchment elevation ($\rho$=0.65, $p$<0.005, Figure 6), which in turn is a major control on other hydro-climatic indices ($\bar{Q}$, $\bar{P}$) and topographic indices ($DD$, $G$, $L$, $L/G$ and $TWI$).

Across the 22 catchments, $F_{yw}^*$ is positively correlated with the areal fraction of saturated soils ($\rho$=0.58,

$p$<0.01) and low-permeability soils ($\rho$=0.52, $p$<0.05). These relationships remain significant when the snow-dominated sites are omitted from the analysis. A strong positive relationship with $F_{yw}^*$ can be expected because saturated and low-permeability soils are often associated with overland flow and fast subsurface runoff mechanisms triggered by precipitation intensity (infiltration excess) or exceedance of soil water storage thresholds (saturation excess) (Dunne and Black, 1970; Horton, 1933). Particularly

high fractions of saturated soils occur at three small neighbouring catchments (Erlenbach, Lümpenenbach and Vogelbach), which are characterized by shallow gleysols (Feyen et al., 1996; Fischer et al., 2015). Together with the nearby Biber catchment, these four sites exhibit the largest young water fractions in our data set. $F_{yw}^*$ is not significantly correlated with the areal fractions mapped as having high, intermediate, or low groundwater productivity, here used as a proxy for the catchments'

hydrogeologic properties. This result is perhaps unsurprising; most groundwater is probably older than the threshold age that defines young water, so the young water fraction will not be sensitive to how much older the groundwater is. Instead, the fraction of young water should primarily reflect processes that control flow processes and routing near the land surface (shallow groundwater, soil water, overland flow) rather than the groundwater flow in deep aquifers where flow velocities can be several orders of

magnitude slower.

The young water fraction is strongly correlated with the areal fraction of forest ($\rho$=0.58, $p$<0.01) across our study catchments (Table 1, Table 4). Excluding the snow-dominated sites from the analysis slightly weakens this relationship although it remains statistically significant ($\rho$=0.51, $p$<0.05). One would normally expect tree roots to increase soil permeability, resulting in greater infiltration and groundwater

recharge (Brantley et al., 2017). Thus the correlation we observe may be artefactual, since across our sites forest cover is also correlated with higher drainage densities and shorter mean flow paths, as well as higher fractions of saturated and low-permeability soils, all of which can plausibly increase the young water fraction. More generally, among our 22 study sites, hydro-climatic characteristics are correlated with landscape properties, making it challenging to clearly identify individual controls on the

young water fraction. Broadly, however, we can conclude that high young water fractions are generally associated with hydro-climatic factors (e.g., humid climate and infiltration excess) and landscape characteristics (e.g., low soil permeability and dense drainage networks) that facilitate fast streamflow responses.



## 6 Discharge sensitivity of the young water fraction as a diagnostic indicator of runoff generation processes

The catchment inter-comparison analysis presented in Sect. 5 suggests that wetter catchments, and those with shorter and faster flowpaths, have larger young water fractions and bigger seasonal cycles in

streamwater isotopes. In individual catchments, one would also expect young water fractions (and thus seasonal isotope cycles) to be variable in time, i.e., to be larger during periods of stronger precipitation forcing and wetter antecedent conditions, as shallower, faster flow paths become more dominant, and as the stream network extends farther into the landscape, shortening the average path length of subsurface flow (Godsey and Kirchner, 2014). In this section, we examine how strongly young water fractions

respond to changes in catchment wetness, as reflected in stream discharge.

### 6.1 Young water fractions of distinct flow regimes

Our expectation that the young water fraction should be higher under wetter conditions (and thus during higher stream discharges) is borne out by the observation that flow-weighted young water fractions are systematically higher than un-weighted young water fractions (Sect. 4.3). We can visualize the

relationship between $F_{yw}$ and stream discharge (as a proxy for catchment wetness) by separating the streamwater isotope time series into different discharge ranges and calculating the seasonal isotope cycles and $F_{yw}$ values individually for each of these flow regimes. In Figure 7, we show $F_{yw}$ for six distinct flow regimes (the 1st to 4th quartiles, as well as the upper 20 % and 10 %, of daily discharges) at nine of our 22 study sites. In our data set, these sites have the longest isotope time series, allowing us to

estimate robust seasonal cycle coefficients $A_S$ for each individual flow regime.

The visual patterns shown in Figure 7 are similar for catchments located close to each other, such as for Alp and Biber, or for Lümpenenbach, Vogelbach and Erlenbach. However, among the 9 sites, young water fractions vary substantially, with $F_{yw}$ in the lowest flow regime ranging from 0.03 at Dischmabach to 0.29 at Erlenbach and $F_{yw}$ in the highest flow regime ranging from 0.13 at Ilfis to 0.60

at Biber.

Figure 7 suggests that the relationship between discharge and $F_{yw}$ may be a diagnostic fingerprint linked to catchments' hydrological properties that control the storage and release of young water. However, the nine catchments shown in Figure 7 are too small of a sample to draw any robust conclusions concerning how this fingerprint may vary with catchments' landscape characteristics and hydro-climatic

conditions.





### 6.2 Estimating the discharge sensitivity of $F_{yw}$ and linking it to the catchment's landscape and hydro-climatic characteristics

As a first-order estimate of the sensitivity of $F_{yw}$ to discharge across all 22 study catchments, we calculated the linear slope of the relationship between $Q$ and $F_{yw}$, using a method that does not require breaking the streamwater isotope time series into separate flow regimes (and thus has more modest data requirements than plots like Figure 7). For each site, we assume that the seasonal amplitude of precipitation isotopes ($A_P$) is independent of $Q$, leaving the seasonal amplitude of streamwater isotopes $A_S$ as the only flow-rate dependent variable. If $A_S$ varies with discharge but $A_P$ does not, then the young water fraction $F_{yw}$ varies with $Q$ as:

$$F_{yw}(Q) = A_S(Q)/A_P \qquad . \tag{7}$$

If we approximate $A_S$ as a linear function of $Q$,

$$A_S(Q) = n_S + m_S Q \qquad , \tag{8}$$

we can estimate the linear slope ($m_S$) and the intercept ($n_S$) through nonlinear fitting (analytic Gauss-Newton) by replacing $A_S$ in Eq. (2) with $A_S(Q)$ from Eq. (7), yielding:

$$c_S(t) = (n_S + m_S Q) \cdot \sin(2\pi f t - \varphi_S) + k_S \tag{9}$$

In Eq. (9), $\varphi_S$ is the phase of the seasonal cycle (rad), $t$ is the time (decimal years), $f$ is the frequency (years$^{-1}$) and $k_S$ (‰) is a constant describing the vertical offset of the isotope signal. For the sake of simplicity, Eq. (9) assumes that the amplitude of the seasonal cycle varies with $Q$ but the phase $\varphi_S$ does not. Numerical experiments (e.g., Fig. 8 in Kirchner, 2016b) suggest that the change in streamwater isotope cycle phase $\varphi_S$ between high and low flows should have only a minor influence on the estimate of the parameters in Eq. (9), because the change in $\varphi_S$ can only be large when the cycle is strongly damped (i.e., during low-flow conditions), and the phase of such a strongly damped cycle will have little effect on the fit to the data.

Combining Eqs. (7) and (8) yields

$$F_{yw}(Q) = \frac{n_S + m_S Q}{A_P} = \frac{n_S}{A_P} + \frac{m_S}{A_P} Q \tag{10}$$

and thus, the linear slope of the dependence of $F_{yw}$ on $Q$ can be approximated as $m_S/A_P$, which has units of $Q^{-1}$. The uncertainty in this slope was estimated through Gaussian error propagation.

We term this linear slope the "discharge sensitivity" of $F_{yw}$. Our use of this term should not be interpreted to mean that $F_{yw}$ depends, in a mechanistic sense, on discharge per se. Instead, we use the term to indicate the statistical sensitivity of $F_{yw}$ to discharge, where discharge is a proxy indicator of catchment wetness conditions and hydro-climatic forcing. Catchments with high discharge sensitivity of $F_{yw}$ (steep linear slope in Eq. (10)) are ones in which the young water fraction varies greatly between



low and high flows, suggesting that faster flowpaths are more predominant in larger events. Conversely, catchments with low discharge sensitivity (shallower linear slopes in Eq. (10)) are ones in which young water fractions are broadly similar between low and high flows, suggesting that the same predominant flowpaths are activated in similar proportions in both large and small runoff events.

At the Aach catchment, only two streamwater samples were collected during high-flow conditions, resulting in an unrealistic and highly uncertain value for $m_S$. At the remaining 21 sites, the linear slopes of the $Q$-$F_{yw}$ relationships range between zero (within error) at Ilfis and Sitter, and 0.0732±0.0360 d mm$^{-1}$ at Mentue, with an average value of 0.0202±0.0046 d mm$^{-1}$.

There was no systematic relationship between the young water fraction (either $F_{yw}$ or $F_{yw}^*$) and the

discharge sensitivity, indicating that they are different and largely independent measures of catchment behaviour (Figure 8 and Figure 9).  The discharge sensitivity of $F_{yw}$ is, however, strongly correlated to a range of landscape and hydro-climatic conditions, including $\bar{P}$ ($\rho$=0.64, see also Figure 9b), $\bar{P}_{intensity}$ ($\rho$=0.56), $\bar{Q}$ ($\rho$=0.61), $DD$ ($\rho$=0.59), $L/G$ ($\rho$=0.75), $L$ ($\rho$=0.46), $G$ ($\rho$=0.46), $TWI$ ($\rho$=0.52), $A_{precip}$ ($\rho$=0.44), and mean catchment elevation ($\rho$=0.44).  All of these correlations remain statistically

significant (and many become stronger) when the snow-dominated sites are excluded from the analysis.

These results suggest that catchments with low discharge sensitivity of $F_{yw}$ are characterized by high elevations, dense river networks (high $DD$, low $L/G$) and/or generally humid conditions (high $\bar{P}$). These catchment properties are generally associated with predominantly shallow runoff flowpaths during both large and small precipitation events, such that the fraction of young water remains relatively

high under widely varying flow regimes.  In contrast, in catchments characterized by lower drainage density and less humid conditions, larger or higher-intensity storms are likely to strongly alter the proportions of different dominant flowpaths, leading to bigger variations in $F_{yw}$ (i.e., higher discharge sensitivity).  For example, the dynamic extension of the stream network (e.g., Godsey and Kirchner, 2014; Jensen et al., 2017) and/or the increase in hydrologic connectivity between the stream network

and the surrounding landscape (e.g., Detty and McGuire, 2010; Phillips et al., 2011; von Freyberg et al., 2015) should more strongly influence the relative proportion of young streamflow in catchments where drainage density is not already high.  Likewise, the activation of shallow flowpaths during larger storm events will have a bigger influence on $F_{yw}$ in drier catchments than in wetter ones, where shallow flowpaths are likely to be activated during both large and small events.

Equation (10) quantifies discharge sensitivity based on the linear slope of the relationship between $F_{yw}$ and $Q$, whereas Figure 7 shows how $F_{yw}$ varies with log($Q$) for different fractions of the discharge distribution.  By replacing $Q$ with log($Q$) in Eqs. (7)-(10), one could easily determine the linear slope of the relationship between $F_{yw}$ and log($Q$).  However, calculating linear slopes between $F_{yw}$ and log($Q$)



yields no significant correlations with any of the variables in Table 2 or Table S1.  It should be noted that calculations based on $\log(Q)$ will be more strongly influenced by small discharges, whereas calculations based on $Q$ will be more strongly influenced by the upper tail of the $Q$ distribution.  Thus, since our primary focus is storm runoff generation, we calculated the discharge sensitivity based on $Q$

instead of $\log(Q)$, yielding significant correlations with several catchment characteristics as shown in Table 4.

Interestingly, although $F_{yw}$ and its discharge sensitivity are not significantly correlated with each other, they are often correlated with catchment characteristics in opposite ways (Table 4).  For example, high $DD, \bar{Q}, \bar{P}, \bar{P}_{intensity}, QFI$, and $CV_Q$ exhibit statistically significant correlations with higher $F_{yw}$ but also

exhibit statistically significant correlations with lower discharge sensitivity of $F_{yw}$.  In catchments with dense river networks and/or generally humid climates, fast runoff flowpaths will dominate (and thus $F_{yw}$ and $F_{yw}^*$ will be high).  These same conditions should also make fast runoff flowpaths more persistent, with the result that the young water fraction will not be strongly dependent on catchment wetness conditions or hydro-climatic forcing (and thus discharge sensitivity will be low).

**6.3   A conceptual model of the mechanistic relationship between young water fractions and discharge**

Figure 10 presents a conceptual summary of the relationships between the young water fraction, its discharge sensitivity, and landscape and hydro-climatic characteristics that control streamflow generation.  We suggest that the general trend of the $Q$-$F_{yw}$ relationship is positive because high-flow

periods during precipitation events are likely to contain larger fractions of young water traveling by quick flow paths, while low-flow conditions are primarily sustained by older groundwater.  In Figure 10, the steepness of the linear slope expresses how extensively fast flowpaths are activated during high flows.  In theory, a linear slope of zero (i.e., $F_{yw}$ insensitive to discharge) would represent strictly linear rainfall-runoff behaviour with a constant mixing fraction of young and old water.  In natural systems,

however, the relative proportions of streamflow generation mechanisms are likely to vary between high and low flows, making $F_{yw}$ sensitive to discharge.  Low discharge sensitivities of $F_{yw}$ can occur at sites with either high or low young water fractions (cases 1 and 3, respectively, in Figure 10 ; e.g., Erlenbach and Ilfis, respectively, in Figure 7).  Case 1 might be found in humid catchments with frequent precipitation, low storage capacity and dense river networks, where shallow runoff flowpaths dominate

both during and between events (e.g., triggered by saturation excess).  Case 3 is more likely to occur in catchments with high infiltration capacity and large subsurface storage, where slow subsurface flowpaths dominate both during events and between them, leading to consistently low young water fractions.  A steep linear slope (case 2 in Figure 10 ; e.g., Alp, Biber or Murg in Figure 7) is likely to occur in catchments where the relative contributions of fast and slow flowpaths vary dramatically in





response to hydro-climatic forcing or antecedent wetness conditions. Such variations in the relative dominance of fast and slow flowpaths should be more common in less humid catchments where, dependent on antecedent wetness conditions, variations in precipitation amount and intensity trigger drainage network expansion, as well as shifts in hydrological connectivity due to groundwater tables

rising into more permeable layers.

The hydrological concepts presented in Figure 10 are based on the young water fraction analysis for 21 Swiss catchments that share some common landscape and hydro-climatic characteristics, such as similar vegetation cover, relatively humid climate, and (partly) mountainous terrain. Hence, we must be cautious about extending this conceptual model to regions characterized by (semi-) arid or arctic

climates, very different vegetation cover or predominantly flat terrain. In addition, linking young water fractions to catchment wetness conditions and hydro-climatic forcing may be difficult in catchments with streamflow regimes that are discontinuous or strongly affected by large proportions of lakes, water management (e.g., groundwater pumping, artificial groundwater recharge, irrigation or water diversion) or land-use change (e.g., urban development, soil degradation, or forest clear cutting). Nevertheless,

long-term tracer data sets from other catchments could be used to expand our analysis beyond the Swiss study sites and to test the transferability of the conceptual model presented in Figure 10.

## 7   Summary and Conclusions

The fraction of streamflow younger than roughly 2-3 months has recently been proposed as a robust measure of water age in heterogeneous catchments, which can be calculated directly from the seasonal

cycles of stable water isotopes in precipitation and streamflow (Kirchner, 2016a, b). Here, we have leveraged an extensive isotope data set from 22 small- to medium-sized Swiss catchments to explore how the young water fraction ($F_{yw}$) varies with catchment characteristics and climatic forcing.

Catchment inter-comparison studies require applying consistent procedures across sites, so we quantified how choices of methodology may affect estimates of $F_{yw}$. Across the 22 sites, $F_{yw}$ values

were not particularly sensitive to the spatial interpolation methods used to estimate precipitation isotope signatures (Sect. 4.1), or sensitive to whether one accounts for snow accumulation and melt in estimating isotopic inputs to the catchment (Sect. 4.2). Flow-weighting the streamwater isotope measurements, however, yielded flow-weighted young water fractions ($F_{yw}^{*}$) that were roughly 26 % larger than their un-weighted counterparts ($F_{yw}$; Sect. 4.3, Figure 5). This result is not surprising,

because flow peaks typically follow intense rainfall and thus should contain more recent precipitation than base flows. However, our results quantify, for the first time, how flow-weighting affects young water fractions using real-world data.





The flow-weighted young water fractions of the 22 Swiss catchments ranged from 0.07±0.01 to
0.49±0.03 (±$SE$), whereas the un-weighted $F_{yw}$ were slightly smaller, ranging from 0.06±0.01 to
0.37±0.03.  The (unweighted) $F_{yw}$ values from our Swiss catchments span roughly the 10[th] to 80[th]
percentiles of the $F_{yw}$ values estimated by Jasechko et al. (2016) for 254 rivers around the world.  The
median $F_{yw}$ among the 22 Swiss catchments was 0.16 (95 % confidence interval 0.10 – 0.21), somewhat
less than the global median of 0.21 (95 % confidence interval 0.19-0.24; Jasechko et al., 2016),
consistent with Jasechko et al.'s observation that young water fractions tend to be lower in steeper
landscapes.  Among the 22 Swiss catchments, $F_{yw}$ and $F_{yw}^*$ were positively correlated with catchment
characteristics that control wetness conditions (e.g., mean monthly precipitation, and mean precipitation
intensity) and near-surface flow routing (e.g., drainage density and areal fractions of saturated soils; see
Sect. 5).

By calculating young water fractions for individual ranges of streamflow, we demonstrated that young
water fractions generally increase with discharge, and that this sensitivity of $F_{yw}$ to discharge varies
from site to site (Sect. 6.1, Figure 8).  We developed a method to quantify the discharge sensitivity of
$F_{yw}$ through calculating the linear slope of the $Q$-$F_{yw}$ relationship (Eqs. (7) to (10)).  The discharge
sensitivity expresses how $F_{yw}$ responds to changes in river discharge, which is used here as a proxy for
catchment wetness and hydro-climatic forcing.  For our study catchments, the young water fraction and
its discharge sensitivity were not correlated with each other, suggesting that these metrics represent
different diagnostic indicators of catchment hydrologic behaviour (Sect. 6.2, Figure 8).  We found that
low discharge sensitivities imply greater persistence in the relative contributions of fast and slow
flowpaths to streamflow during both high and low flows.  High discharge sensitivities, on the other
hand, imply shifts in flowpath dominance during higher flows, such as when subsurface water tables
rise into more permeable layers or the river network expands further into the landscape.

Based on our analysis, we developed a generalized conceptual description that relates $F_{yw}$ and its
discharge sensitivity to dominant streamflow generation mechanisms (Sect. 6.3, Figure 10).  It remains
to be tested whether this conceptual description is transferable to other sites with landscape features and
hydro-climatic forcing that are substantially different from our 22 Swiss study catchments.  By
clarifying several methodological issues in estimating young water fractions, and by presenting a
theoretical framework for quantifying discharge sensitivity of $F_{yw}$, we hope to contribute to future
catchment (inter-comparison) studies.

**Acknowledgments**

The collection and analysis of the streamwater isotope data were mainly funded as part of the National
Research Programme NRP 61 by the Swiss National Science Foundation within the project Drought-





CH. We thank Massimiliano Zappa from the Swiss Federal Research Institute WSL for providing interpolated meteorological data for the 22 study catchments and Wouter Berghuijs for helpful discussions.

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





## Tables

Table 1: General properties of the 22 study catchments and streamwater isotope time series

| Catchment name | Gauging station | Longitude (WGS84) | Latitude (WGS84) | Area (km²) | Mean elevation (from-to) (m) | Average annual precipitation [a] (mm) | Hydro-climatic regime | δ¹⁸O in streamwater from-to (mm/yyyy) (number of samples) |
|---|---|---|---|---|---|---|---|---|
| Aabach | Mönchaltdorf | 8.7206 | 47.3110 | 49.0 | 635 (519-1092) | 1358 | Rainfall dominated | 09/2010-02/2013 (62) |
| Aach | Salmsach, Hungerbühl | 9.3572 | 47.5505 | 50.0 | 472 (408-560) | 1141 | Rainfall dominated | 07/2010-12/2011 (26) |
| Allenbach | Adelboden | 7.5521 | 46.4860 | 28.8 | 1852 (1293-2742) | 1338 | Snow dominated | 09/2010-05/2015 (87) |
| Alp | Einsiedeln | 8.7393 | 47.1508 | 46.5 | 1154 (845-1894) | 1776 | Hybrid | 05/2010-11/2015 (131) |
| Biber | Biberbrugg | 8.7209 | 47.1534 | 31.6 | 999 (827-1495) | 1658 | Rainfall dominated | 05/2010-11/2015 (140) |
| Dischmabach | Davos, Kriegsmatte | 9.8772 | 46.7754 | 43.2 | 2369 (1663-3139) | 1072 | Snow dominated | 10/2010-05/2015 (81) |
| Emme | Eggiwil, Heidbüel | 7.8047 | 46.8711 | 127.0 | 1285 (743-2216) | 1372 | Hybrid | 06/2010-11/2013 (71) |
| Ergolz | Liestal | 7.7342 | 47.4882 | 261.2 | 584 (305-1165) | 1081 | Rainfall dominated | 06/2010-11/2015 (140) |
| Erlenbach | Erlenbach | 8.7089 | 47.0452 | 0.7 | 1359 (1117-1650) | 1853 | Hybrid | 07/2010-05/2015 (140) |
| Guerbe | Belp, Mülimatt | 7.5155 | 46.7888 | 55.4 | 1037 (556-2152) | 1236 | Hybrid | 07/2010-12/2012 (64) |
| Ilfis | Langnau | 7.7975 | 46.9379 | 187.9 | 1037 (681-2087) | 1443 | Hybrid | 07/2010-05/2015 (128) |
| Langeten | Huttwil, Häberenbad | 7.8282 | 47.1225 | 60.3 | 760 (598-1100) | 1297 | Rainfall dominated | 07/2010-05/2015 (103) |
| Lümpenenbach | Lümpenenbach | 8.7052 | 47.0467 | 0.9 | 1336 (1092-1508) | 1803 | Hybrid | 10/2010-11/2015 (132) |
| Mentue | Yvonand, La Mauguettaz | 6.7242 | 46.7768 | 105.0 | 679 (447-926) | 1097 | Rainfall dominated | 07/2010-02/2013 (63) |
| Murg | Waengi | 8.9529 | 47.4963 | 76.8 | 648 (467-1036) | 1314 | Rainfall dominated | 07/2010-11/2014 (95) |
| Ova da Cluozza | Zernez | 10.1183 | 46.6932 | 26.9 | 2364 (1519-3160) | 887 | Snow dominated | 08/2010-09/2013 (65) |
| Riale di Calneggia | Cavergno, Pontit | 8.5429 | 46.3696 | 23.9 | 1986 (881-2908) | 1686 | Snow dominated | 07/2010-12/2012 (55) |
| Rietholzbach | Mosnang, Rietholz | 9.0123 | 47.3761 | 3.2 | 794 (671-938) | 1415 | Rainfall dominated | 07/2010-02/2013 (68) |
| Schaechen | Bürglen, Galgenwäldli | 8.6517 | 46.8710 | 107.9 | 1719 (487-3260) | 1565 | Snow dominated | 04/2011-05/2015 (66) |
| Sense | Thoerishaus, Sensematt | 7.3514 | 46.8883 | 351.2 | 1068 (554-2184) | 1258 | Hybrid | 10/2010-03/2013 (47) |
| Sitter | Appenzell | 9.4104 | 47.3319 | 88.2 | 1301 (768-2500) | 1771 | Hybrid | 11/2010-05/2015 (97) |
| Vogelbach | Vogelbach | 8.7161 | 47.0761 | 1.6 | 1335 (1038-1540) | 1800 | Hybrid | 06/2010-11/2015 (139) |

[a] Based on interpolated data from PREVAH and the time period 01/2000 - 12/2015



**Table 2: Hydro-climatic and topographic indices, as well as soil properties of the 22 study catchments**

| Catchment name | Quick flow index $QFI$ (-) | Coefficient of variation of $Q$ (%) | Average discharge $\bar{Q}$ (mm month⁻¹) | Average precipitation $\bar{P}$ (mm month⁻¹) | Average precip. intensity (mm d⁻¹) | Precipitation amplitude $A_{precip}$ (mm month⁻¹) | Median flow path length $L$ (m) | Median flow path gradient $G$ (m m⁻¹) | $L/G$ (m) | Drainage density $DD$ (km km⁻²) | Topo-graphic gradient (%) | Median topographic wetness index $TWI$ (-) | Fraction forested area (%) | Fraction low-permeab. soils (%) | Fraction saturated soils (%) |
|---|---|---|---|---|---|---|---|---|---|---|---|---|---|---|---|
| Aabach | 0.62 | 126.39 | 54.9 | 106.1 | 5.4 | 0.4 | 407 | 0.04 | 10175 | 1.97 | 5.5 | 9.22 | 14.8 | 0 | 17 |
| Aach | 0.63 | 137.20 | 33.3 | 85.1 | 4.8 | 0.4 | 481 | 0.02 | 24050 | 1.46 | 2.1 | 9.61 | 14.3 | 0 | 13 |
| Allenbach | 0.40 | 100.77 | 109.3 | 99.4 | 4.7 | 1.0 | 423 | 0.31 | 1365 | 2.45 | 45.1 | 9.67 | 15.9 | 57 | 20 |
| Alp | 0.66 | 138.90 | 126.3 | 158.2 | 6.3 | 0.6 | 196 | 0.21 | 933 | 3.88 | 25.7 | 10.92 | 50.6 | 48 | 23 |
| Biber | 0.72 | 149.36 | 96.2 | 150.2 | 5.8 | 0.5 | 207 | 0.16 | 1294 | 3.77 | 18.2 | 9.48 | 43.2 | 30 | 15 |
| Dischmabach | 0.31 | 85.61 | 99.5 | 76.4 | 3.8 | 1.3 | 647 | 0.33 | 1961 | 1.74 | 46.1 | 10.07 | 2.8 | 59 | 0 |
| Emme | 0.72 | 142.06 | 88.7 | 116.6 | 4.8 | 0.6 | 286 | 0.27 | 1059 | 3.38 | 33.1 | 9.76 | 38.8 | 49 | 27 |
| Ergolz | 0.59 | 118.42 | 39.5 | 87.7 | 4.1 | 0.2 | 421 | 0.15 | 2807 | 1.34 | 19.5 | 9.99 | 39.9 | 41 | 1 |
| Erlenbach | 0.81 | 169.73 | 138.9 | 162.4 | 6.6 | 0.8 | 169 | 0.20 | 845 | 6.61 | 23.9 | 9.27 | 62.3 | 4 | 95 |
| Guerbe | 0.49 | 90.33 | 100.3 | 94.9 | 4.3 | 0.5 | 258 | 0.19 | 1358 | 2.59 | 27.6 | 11.67 | 30.6 | 35 | 13 |
| Ilfis | 0.53 | 113.61 | 79.2 | 127.5 | 5.2 | 0.5 | 157 | 0.30 | 523 | 3.31 | 28.6 | 8.87 | 50.0 | 28 | 6 |
| Langeten | 0.30 | 61.72 | 53.9 | 118.2 | 4.7 | 0.3 | 308 | 0.11 | 2800 | 1.70 | 12.6 | 9.52 | 18.4 | 0 | 3 |
| Lümpenenbach | 0.68 | 141.41 | 152.0 | 157.1 | 6.0 | 0.7 | 155 | 0.17 | 912 | 6.57 | 19.6 | 9.85 | 29.3 | 4 | 96 |
| Mentue | 0.52 | 154.74 | 29.7 | 89.3 | 3.9 | 0.2 | 364 | 0.08 | 4550 | 1.47 | 8.9 | 9.10 | 23.0 | 0 | 0 |
| Murg | 0.52 | 97.33 | 62.8 | 116.6 | 5.1 | 0.3 | 219 | 0.10 | 2190 | 2.16 | 12.0 | 10.83 | 32.0 | 0 | 8 |
| Ova da Cluozza | 0.39 | 88.68 | 72.8 | 61.3 | 3.5 | 1.2 | 616 | 0.46 | 1339 | 0.72 | 59.3 | 9.43 | 13.8 | 34 | 0 |
| Riale di Calneggia | 0.52 | 154.05 | 143.7 | 129.3 | 6.4 | 1.1 | 647 | 0.46 | 1407 | 1.03 | 64.4 | 9.51 | 15.3 | 44 | 0 |
| Rietholzbach | 0.69 | 140.73 | 87.2 | 121.1 | 5.6 | 0.4 | 194 | 0.18 | 1078 | 2.09 | 14.6 | 9.88 | 22.3 | 0 | 0 |
| Schaechen | 0.40 | 86.35 | 120.3 | 140.0 | 5.9 | 0.9 | 646 | 0.38 | 1700 | 1.36 | 54.3 | 9.38 | 19.2 | 67 | 5 |
| Sense | 0.53 | 101.76 | 54.2 | 95.2 | 4.3 | 0.5 | 227 | 0.20 | 1135 | 2.84 | 24.1 | 9.00 | 32.9 | 24 | 19 |
| Sitter | 0.62 | 120.79 | 94.2 | 168.7 | 6.4 | 0.6 | 329 | 0.27 | 1219 | 2.65 | 35.2 | 9.96 | 30.8 | 61 | 14 |
| Vogelbach | 0.70 | 162.67 | 117.3 | 162.2 | 6.3 | 0.8 | 193 | 0.28 | 689 | 6.55 | 28.9 | 9.69 | 82.1 | 51 | 49 |




**Table 3: Values±standard errors of flow-weighted seasonal amplitude coefficients of precipitation isotopes ($A_P$), unweighted and flow-weighted seasonal amplitude coefficients of streamwater isotopes ($A_S$), unweighted and flow-weighted young water fractions, as well as the discharge sensitivity of the young water fraction (estimated as the linear slope of the $Q$–$F_{yw}$–relationship; see Sect. 6).**

| Catchment name | $A_P \pm SE$ (‰) | $A_S \pm SE$ (‰) | | $F_{yw} \pm SE$ (-) | | Discharge sensitivity of |
|---|---|---|---|---|---|---|
| | Flow-weighted | Unweighted | Flow-weighted | Unweighted | Flow-weighted | $F_{yw} \pm SE$ (d/mm) |
| Aabach | 3.57±0.18 | 0.55±0.09 | 0.77±0.12 | 0.15±0.03 | 0.22±0.04 | 0.0530±0.0247 |
| Aach | 3.65±0.18 | 0.57±0.12 | 0.35±0.16 | 0.16±0.03 | 0.10±0.04 | – [a] |
| Allenbach | 5.54±0.27 | 0.48±0.06 | 0.61±0.08 | 0.09±0.01 | 0.11±0.02 | 0.0185±0.0065 |
| Alp | 3.50±0.19 | 0.97±0.07 | 1.24±0.08 | 0.28±0.03 | 0.35±0.03 | 0.0119±0.0063 |
| Biber | 3.39±0.19 | 0.86±0.07 | 1.33±0.10 | 0.25±0.03 | 0.39±0.04 | 0.0299±0.0074 |
| Dischma | 6.36±0.29 | 0.46±0.04 | 0.66±0.05 | 0.07±0.01 | 0.10±0.01 | 0.0169±0.0021 |
| Emme | 3.80±0.20 | 0.88±0.10 | 1.22±0.11 | 0.23±0.03 | 0.32±0.03 | 0.0237±0.0107 |
| Ergolz | 3.15±0.18 | 0.30±0.04 | 0.43±0.06 | 0.09±0.01 | 0.14±0.02 | 0.0651±0.0186 |
| Erlenbach | 4.63±0.22 | 1.74±0.09 | 2.27±0.10 | 0.37±0.03 | 0.49±0.03 | 0.0066±0.0029 |
| Guerbe | 3.63±0.19 | 0.55±0.07 | 0.67±0.09 | 0.15±0.02 | 0.18±0.03 | 0.0214±0.0113 |
| Ilfis | 3.63±0.19 | 0.40±0.05 | 0.45±0.05 | 0.11±0.01 | 0.12±0.02 | 0.0061±0.0067 |
| Langeten | 3.36±0.18 | 0.20±0.03 | 0.24±0.04 | 0.06±0.01 | 0.07±0.01 | 0.0503±0.0166 |
| Lümpenenbach | 4.66±0.22 | 1.16±0.09 | 1.56±0.10 | 0.25±0.02 | 0.33±0.03 | 0.0174±0.0049 |
| Mentue | 2.72±0.16 | 0.48±0.07 | 0.72±0.09 | 0.18±0.03 | 0.26±0.04 | 0.0732±0.0360 |
| Murg | 3.39±0.18 | 0.27±0.05 | 0.45±0.08 | 0.08±0.01 | 0.13±0.03 | 0.0304±0.0114 |
| Ova da Cluozza | 6.60±0.30 | 0.68±0.09 | 0.91±0.11 | 0.10±0.01 | 0.14±0.02 | 0.0328±0.0077 |
| Riale di Calneggia | 3.94±0.20 | 0.77±0.07 | 0.85±0.12 | 0.20±0.02 | 0.22±0.03 | 0.0451±0.0145 |
| Rietholzbach | 3.54±0.18 | 0.43±0.04 | 0.71±0.06 | 0.12±0.01 | 0.20±0.02 | 0.0132±0.0045 |
| Schaechen | 3.67±0.16 | 0.58±0.05 | 0.66±0.07 | 0.16±0.02 | 0.18±0.02 | 0.0174±0.0050 |
| Sense | 3.32±0.19 | 0.61±0.07 | 0.98±0.17 | 0.18±0.02 | 0.29±0.05 | 0.0463±0.0105 |
| Sitter | 3.75±0.18 | 0.74±0.06 | 0.69±0.07 | 0.20±0.02 | 0.19±0.02 | -0.0085±0.0090 |
| Vogelbach | 4.64±0.22 | 1.00±0.06 | 1.42±0.08 | 0.21±0.02 | 0.31±0.02 | 0.0107±0.0034 |

[a] The catchment Aach was omitted from the analysis because its isotope data set contained only two data points during high-flow conditions.





**Table 4:** Spearman rank correlation coefficients relating the flow-weighted ($F^*_{yw}$) and unweighted ($F_{yw}$) young water fractions, and the discharge sensitivity of $F_{yw}$, to selected hydro-climatic indices and landscape properties of the 22 Swiss catchments. The corresponding $p$-values are indicated by regular font in grey fields ($p<0.05$), bold font in grey fields ($p<0.01$), as well as italic and underlined font in grey fields ($p<0.001$); fields without grey shading indicate $p>0.05$.

| | | | Discharge sensitivity[a] | Hydro-climatic indices | | | | | | Topographic indices | | | | | | | | Soils | | |
|---|---|---|---|---|---|---|---|---|---|---|---|---|---|---|---|---|---|---|---|---|
| | $F^*_{yw}$ | $F_{yw}$ | | $\bar{Q}$ | $\bar{P}$ | $A_{precip}$ | Mean $P$ intensity | $QFI$ | $CV_Q$ | Catchm. area | Mean Elevation | Topo. gradient | $DD$ | $TWI$ | $L$ | $G$ | $L/G$ | Fraction forested area | Fraction low-permeab. soils | Fraction saturated soils |
| Flow-weighted young water fraction $F^*_{yw}$ | | 0.90 | -0.14 | 0.44 | 0.55 | 0.14 | 0.52 | 0.73 | 0.75 | -0.30 | 0.13 | -0.03 | 0.64 | 0.01 | -0.52 | -0.02 | -0.55 | 0.58 | 0.52 | 0.58 |
| Unweighted young water fraction $F_{yw}$ | 0.90 | | -0.32 | 0.50 | 0.65 | 0.23 | 0.67 | 0.76 | 0.77 | -0.26 | 0.17 | 0.04 | 0.62 | -0.04 | -0.43 | 0.04 | -0.54 | 0.54 | 0.43 | 0.63 |
| Discharge sensitivity of $F_{yw}$ [a] | -0.14 | -0.32 | | -0.61 | -0.64 | -0.44 | -0.56 | -0.33 | -0.13 | 0.32 | -0.44 | -0.38 | -0.59 | 0.52 | 0.46 | -0.46 | 0.75 | -0.38 | -0.38 | -0.33 |
| Average discharge $\bar{Q}$ | 0.44 | 0.50 | -0.61 | | 0.62 | 0.75 | 0.52 | 0.20 | 0.25 | -0.60 | 0.72 | 0.58 | 0.46 | -0.52 | -0.21 | 0.58 | -0.57 | 0.20 | 0.32 | 0.38 |
| Average precipitation $\bar{P}$ | 0.55 | 0.65 | -0.64 | 0.62 | | 0.20 | 0.91 | 0.52 | 0.47 | -0.32 | 0.20 | 0.11 | 0.64 | -0.27 | -0.57 | 0.15 | -0.62 | 0.58 | 0.49 | 0.50 |
| Precip. amplitude $A_{precip}$ | 0.14 | 0.23 | -0.44 | 0.75 | 0.20 | | 0.26 | -0.11 | -0.03 | -0.50 | 0.95 | 0.84 | 0.11 | -0.70 | 0.25 | 0.83 | -0.36 | -0.18 | -0.08 | 0.13 |
| Mean $P$ intensity | 0.52 | 0.67 | -0.56 | 0.52 | 0.91 | 0.26 | | 0.56 | 0.56 | -0.43 | 0.18 | 0.09 | 0.48 | -0.21 | -0.40 | 0.12 | -0.47 | 0.40 | 0.44 | 0.45 |
| Quickflow index $QFI$ | 0.73 | 0.76 | -0.33 | 0.20 | 0.52 | -0.11 | 0.56 | | 0.82 | -0.27 | -0.20 | -0.30 | 0.66 | 0.11 | -0.62 | -0.26 | -0.51 | 0.62 | 0.70 | 0.62 |
| Coeff. of variation $CV_Q$ | 0.75 | 0.77 | -0.13 | 0.25 | 0.47 | -0.03 | 0.56 | 0.82 | | -0.39 | -0.06 | -0.20 | 0.44 | 0.04 | -0.42 | -0.16 | -0.36 | 0.47 | 0.46 | 0.41 |
| Catchment area | -0.30 | -0.26 | 0.32 | -0.60 | -0.32 | -0.50 | -0.43 | -0.27 | -0.39 | | -0.43 | -0.11 | -0.24 | 0.25 | 0.19 | -0.17 | 0.29 | 0.12 | -0.26 | -0.18 |
| Mean Elevation | 0.13 | 0.17 | -0.44 | 0.72 | 0.20 | 0.95 | 0.18 | -0.20 | -0.06 | -0.43 | | 0.88 | 0.10 | -0.79 | 0.19 | 0.89 | -0.43 | -0.15 | -0.10 | 0.05 |
| Topographic gradient | -0.03 | 0.04 | -0.38 | 0.58 | 0.11 | 0.84 | 0.09 | -0.30 | -0.20 | -0.11 | 0.88 | | -0.07 | -0.86 | 0.33 | 0.97 | -0.37 | -0.09 | -0.19 | -0.11 |
| Drainage density $DD$ | 0.64 | 0.62 | -0.59 | 0.46 | 0.64 | 0.11 | 0.48 | 0.66 | 0.44 | -0.24 | 0.10 | -0.07 | | -0.06 | -0.84 | -0.04 | -0.77 | 0.73 | 0.63 | 0.83 |
| Topogr. wetness index $TWI$ | 0.01 | -0.04 | 0.52 | -0.52 | -0.27 | -0.70 | -0.21 | 0.11 | 0.04 | 0.25 | -0.79 | -0.86 | -0.06 | | -0.02 | -0.91 | 0.59 | -0.02 | 0.03 | 0.07 |
| Flow path length $L$ | -0.52 | -0.43 | 0.46 | -0.21 | -0.57 | 0.25 | -0.40 | -0.62 | -0.42 | 0.19 | 0.19 | 0.33 | -0.84 | -0.02 | | 0.25 | 0.73 | -0.75 | -0.65 | -0.57 |
| Flow gradient $G$ | -0.02 | 0.04 | -0.46 | 0.58 | 0.15 | 0.83 | 0.12 | -0.26 | -0.16 | -0.17 | 0.89 | 0.97 | -0.04 | -0.91 | 0.25 | | -0.45 | -0.06 | -0.16 | -0.13 |
| $L/G$ | -0.55 | -0.54 | 0.75 | -0.57 | -0.62 | -0.36 | -0.47 | -0.51 | -0.36 | 0.29 | -0.43 | -0.37 | -0.77 | 0.59 | 0.73 | -0.45 | | -0.65 | -0.51 | -0.50 |
| Fraction forested area | 0.58 | 0.54 | -0.38 | 0.20 | 0.58 | -0.18 | 0.40 | 0.62 | 0.47 | 0.12 | -0.15 | -0.09 | 0.73 | -0.02 | -0.75 | -0.06 | -0.65 | | 0.66 | 0.55 |
| Fraction low-permeab. soils | 0.52 | 0.43 | -0.38 | 0.32 | 0.49 | -0.08 | 0.44 | 0.70 | 0.46 | -0.26 | -0.10 | -0.19 | 0.63 | 0.03 | -0.65 | -0.16 | -0.51 | 0.66 | | 0.60 |
| Fraction saturated soils | 0.58 | 0.63 | -0.33 | 0.38 | 0.50 | 0.13 | 0.45 | 0.62 | 0.41 | -0.18 | 0.05 | -0.11 | 0.83 | 0.07 | -0.57 | -0.13 | -0.50 | 0.55 | 0.60 | |

a) The catchment Aach was omitted from the analysis because its isotope data set contained only two data points during high-flow conditions.





# Figures

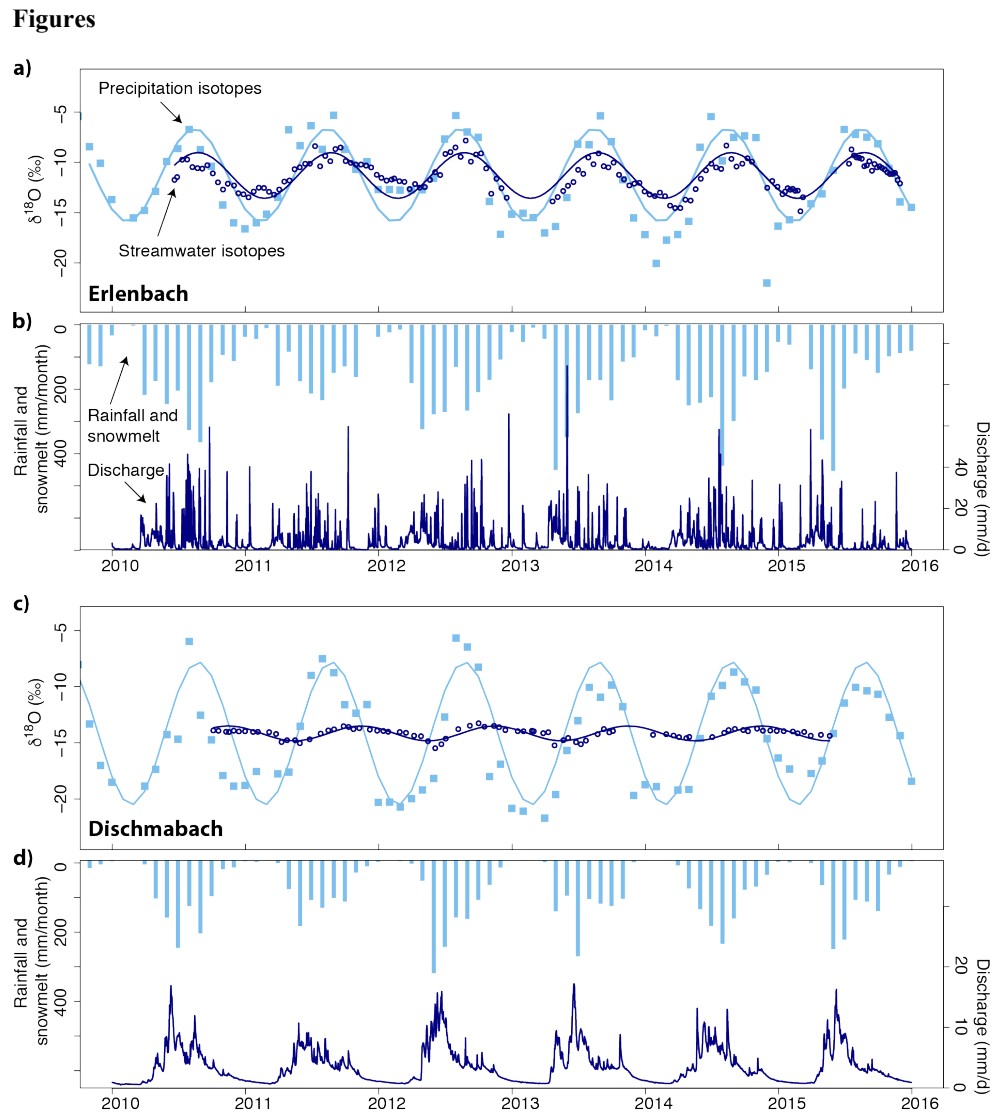

**Figure 1: Hydrologic and isotopic seasonality of precipitation and streamflow for the Erlenbach and Dischmabach catchments.**
**Precipitation isotopes were interpolated with the method of Seeger and Weiler, 2014. Sinusoidal cycles were fitted to the isotope**
5 **data using iteratively reweighted least squares regression. The seasonal cycles of the streamwater isotopes exhibit damping and**
**phase shifts relative to the precipitation isotopic cycles. Stronger damping of the seasonal isotope cycle can be observed in the**
**Dischmabach catchment.**





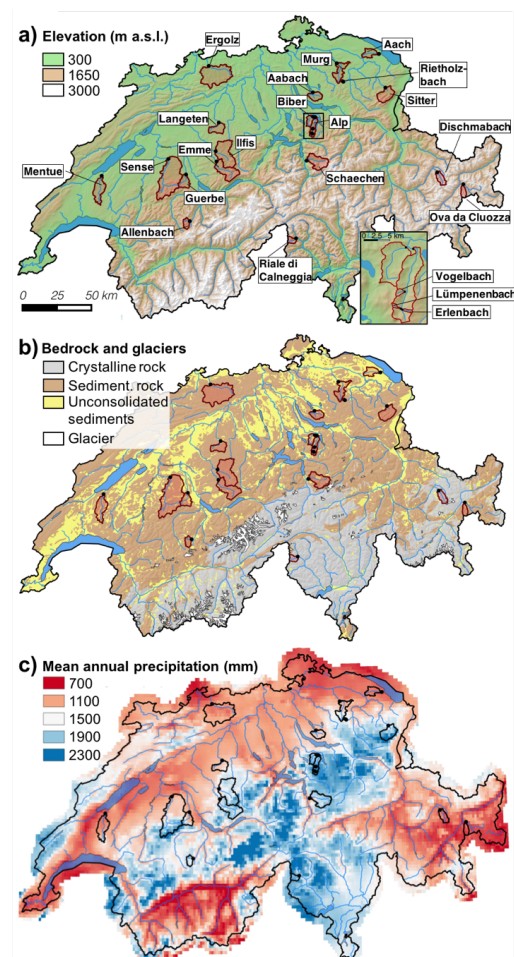

Figure 2: Locations of the 22 study catchments in Switzerland (a), bedrock geology (b), and mean annual precipitation based on the observation period 1961-1990 (c). The expanded panel in (a) shows the sub-catchments of the Alp basin.





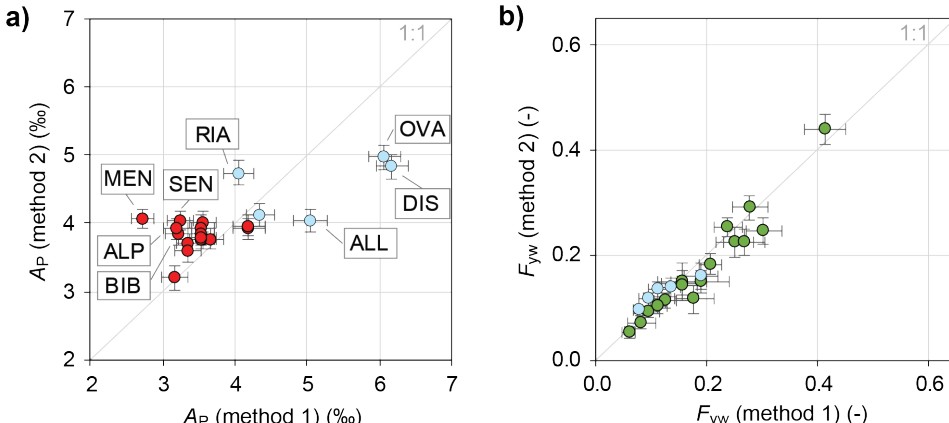

**Figure 3: a) Comparison of flow-weighted seasonal amplitudes of precipitation δ¹⁸O cycles ($A_P$) obtained with two different interpolation methods 1 and 2, those of Seeger and Weiler (2014) and those presented in the Supplement, respectively. Differences in $A_P$ between the two interpolation methods were significant for the catchments highlighted with their abbreviated names. The abbreviations for the study sites stand for Allenbach (ALL), Alp (ALP), Biber (BIB), Dischmabach (DIS), Mentue (MEN), Ova da Clouzza (OVA), Ria di Calneggia (RIA), and Sense (SEN). b) Comparison of young water fractions derived from the two interpolation methods. High-elevation, snow-dominated catchments are marked in light blue colour. Error bars show ±1 standard error.**





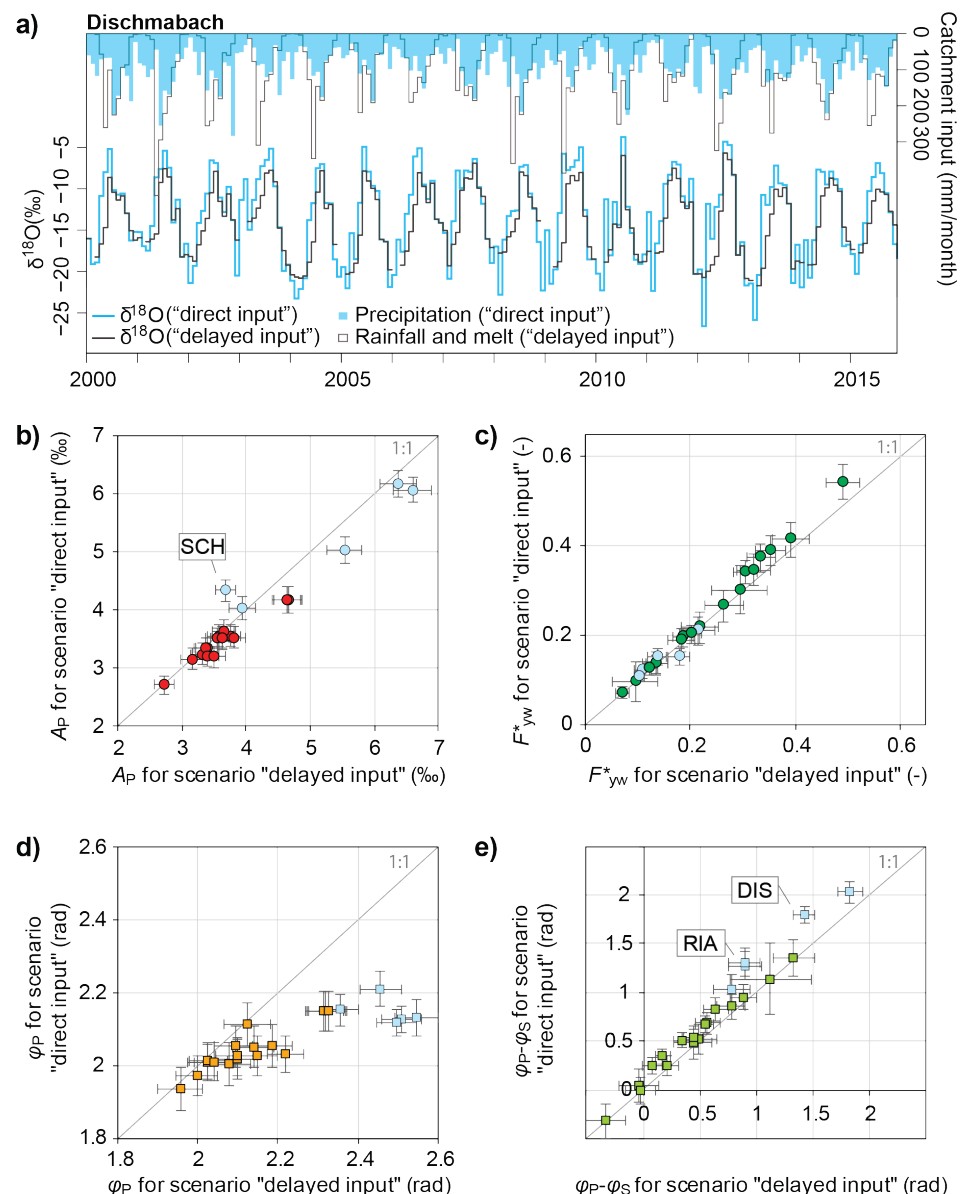

**Figure 4: a)** Time series of catchment input volumes and **δ¹⁸O** values (not volume-weighted) for the Dischmabach catchment calculated using the interpolation method of Seeger and Weiler (2014), with and without modelling of snow accumulation and melt ("delayed input" and "direct input", respectively). Panels b) and c) show the seasonal amplitudes of the precipitation isotope cycles (volume-weighted), and the resulting flow-weighted young water fractions, with and without modelling of snow accumulation and melt ("delayed input" and "direct input", respectively). Panels d) and e) show phases of seasonal precipitation isotope cycles, and the resulting phase shifts, with and without modelling of snow accumulation and melt. High-elevation, snowmelt-dominated sites are marked in light blue. The abbreviations for the study sites stand for Dischmabach (DIS), Ria di Calneggia (RIA), and Schaechen (SCH). Error bars show ±1 standard error.



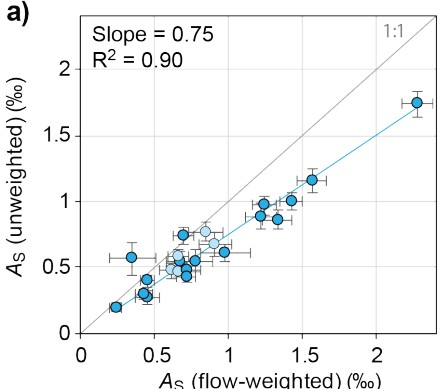

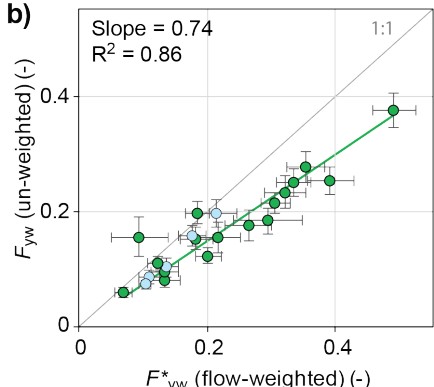

**Figure 5:** Panel a) compares the seasonal amplitudes of streamwater isotope cycles ($A_S$) with and without flow weighting. High-elevation, snowmelt-dominated sites are marked in light blue. Panel b) compares flow-weighted young water fractions ($F^*_{yw}$) with un-weighted young water fractions ($F_{yw}$). Error bars show ±1 standard error. Unweighted young water fractions are roughly 26% smaller than flow-weighted young water fractions across these catchments.



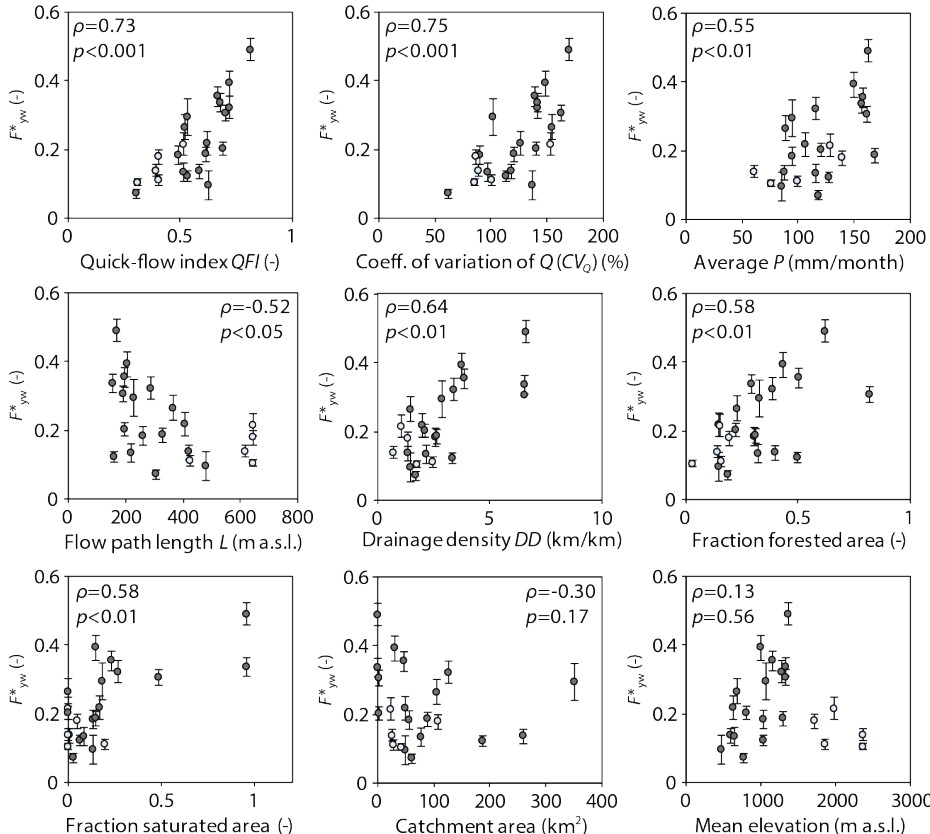

**Figure 6:** Scatterplots showing how young water fractions correlate with climatic and landscape indices. High-elevation, snowmelt-dominated sites are marked in light blue. Error bars show ±1 standard error. Spearman rank correlation coefficients ($\rho$) and corresponding $p$-values are provided in the individual figures.





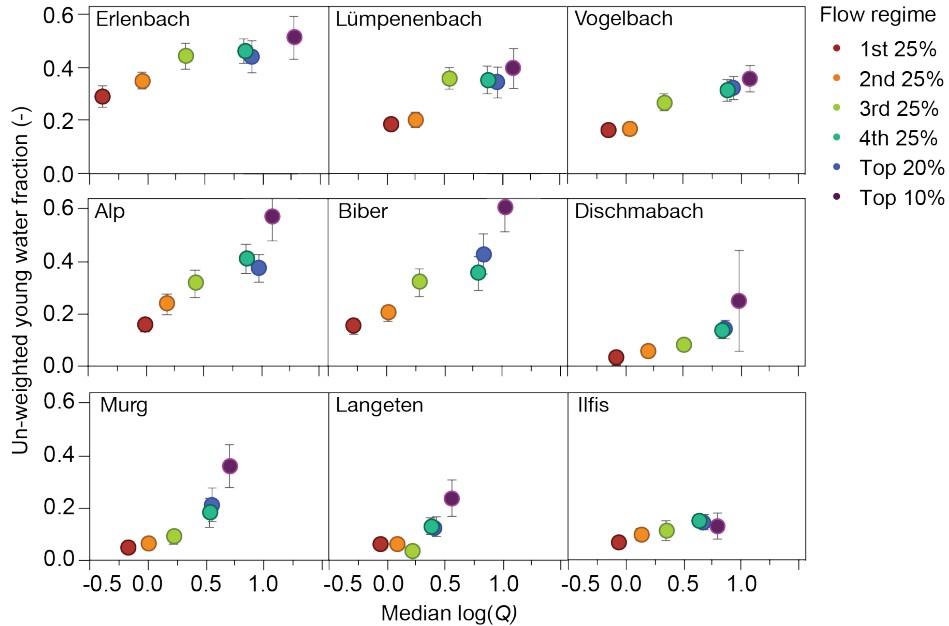

**Figure 7:  Variation in un-weighted young water fractions with flow regime (log-transformed) for the nine Swiss catchments that have sufficiently long time series of streamwater isotope measurements.  Error bars show ±1 standard error.  The young water fraction increases with discharge differently at different sites, suggesting different degrees of activation of fast flowpaths at high flows.**

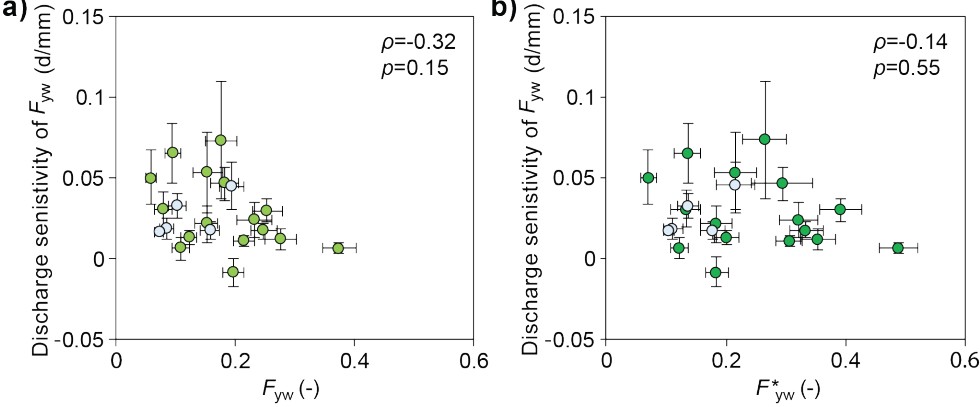

**Figure 8: Scatterplots of the unweighted and flow-weighted young water fractions versus the discharge sensitivity of $F_{yw}$ calculated for 21 of the 22 Swiss catchments (no discharge sensitivity was calculated for the Aach catchment because only two isotope values existed for high-flow conditions).  High-elevation, snowmelt-dominated sites are marked in light blue.  Error bars show ±1 standard error.  There is no systematic relationship between the young water fractions and their discharge sensitivities.**



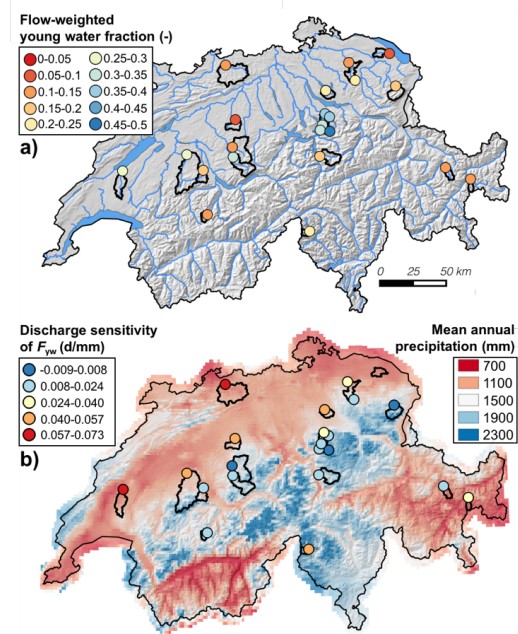

**Figure 9: a)** Flow-weighted young water fractions at the 22 Swiss study catchments; **b)** Discharge sensitivity of $F_{yw}$ at the same sites (mean annual precipitation is shown for comparison).

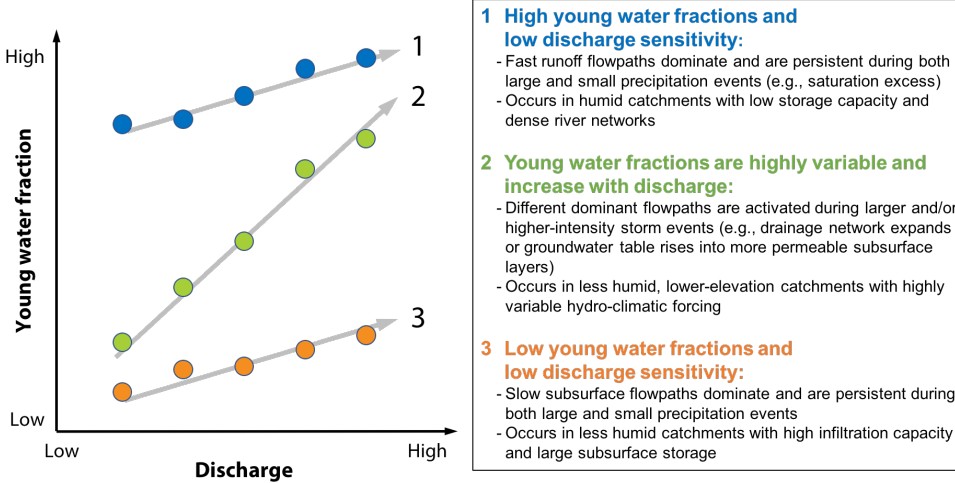

**1 High young water fractions and low discharge sensitivity:**
- Fast runoff flowpaths dominate and are persistent during both large and small precipitation events (e.g., saturation excess)
- Occurs in humid catchments with low storage capacity and dense river networks

**2 Young water fractions are highly variable and increase with discharge:**
- Different dominant flowpaths are activated during larger and/or higher-intensity storm events (e.g., drainage network expands or groundwater table rises into more permeable subsurface layers)
- Occurs in less humid, lower-elevation catchments with highly variable hydro-climatic forcing

**3 Low young water fractions and low discharge sensitivity:**
- Slow subsurface flowpaths dominate and are persistent during both large and small precipitation events
- Occurs in less humid catchments with high infiltration capacity and large subsurface storage

**Figure 10: Conceptual description of the mechanistic relationship between young water fractions and discharge, which is used here as a proxy for catchment wetness and hydro-climatic forcing. The three colours of data points represent three individual hypothetical catchments.**