# Peer review of "Sensitivity of young water fractions to hydro-climatic forcing and landscape properties across 22 Swiss catchments"

_Hydrology and Earth System Sciences, 2017_

## Referee Comment (RC1) · Anonymous Referee #1 · 10 Jan 2018

General comments

This discussion paper presents an analysis of young water fractions (Fyw) in contrasting catchments across Switzerland. The paper first examines the influence of interpolation methods, flow-weighting of measurements and snow in the calculation of Fyw. The second part studies correlations between young water fractions and catchment characteristics. The authors then introduce a new metric (i.e., the discharge sensitivity of Fyw) and relate this metric to catchment characteristics. The paper concludes with a conceptualization of the relationship between young water fractions and streamflow. The paper is well written and addresses important problems in the analysis of isotope data, i.e., interpolation, impact of snow and flow-weighting. Moreover, the paper presents a new concept derived from the recently introduced young water fractions. However, there are some parts that need clarification and rearranging especially in the theoretical and methodological sections.

Specific comments

*Abstract

- Page 1, line 23: suggest clarifying what "this relationship" is (i.e., the relationship between flow and young water fractions).

*Theoretical background

- Page 5, lines 17-20: please clarify that equations (3) and (4) follow from (1) and (2).

- Page 5, lines 25-27: it is not entirely clear what is meant with volume-weighting. I assume it does not refer to the isotope values themselves (so volume-weighting over several samples to obtain a weighted catchment average, as done for the precipitation isotope values), but to the weighting scheme within the IRLS algorithm.

*Data set

- Page 8, lines 6-10: suggest dropping the German terms of the soil properties as this will not mean anything to most readers.

*Results/Discussion

- Due to the concise description of the interpolation methods in the main next, it is not easy for the reader to follow the different steps of the two interpolation methods, although this would be helpful to better understand the differences between the two methods. Moreover, method 2 has been developed by the authors, so this method should be introduced more extensively in the main text. I would thus suggest restructuring the paper by moving major parts of the methodology description from the Supplement to the main text. This could be placed into a subsection of section 3 or a

separate methodological section. Please also explain method 2 in a bit more detail – in the main text, this method is described with one long sentence only. The comparison between the two methods can be kept in section 4.1, which would be more consistent with presenting results only in section 4.

- Page 8, line 26: are these cumulative monthly d18O-values in precipitation (so sampling bottle emptied each month)?

- Page 9, line 25; page 11, line 5 and line 16: "statistically (in)significant" using which statistical method?

- Page 11, line 25: this is the first time the authors mention "gamma distributions". Please clarify that this refers to the underlying transit time distribution model.

- Page 12, lines 29-31: suggest weakening this statement ("consistent with …") as results from a global analysis should be compared with caution to regional analyses and the smaller Fyw in this study could also be caused by various factors other than the gradient dependence. See also page 19, lines 7-8.

- Page 14, lines 15-20: please give a bit more details on the procedure: how many measurements were on average available in each sine-wave regression after separation by flow regimes? Was the number of values sufficient to obtain reliable results? I would expect the seasonal variations to be small and potentially indiscernible under low-flow conditions, when streamflow is dominated by the well-mixed signal of slow flowpaths.

- Page 15, line 3 – page 16, line 4: suggest introducing the concept of discharge sensitivity earlier in the manuscript as a methodological (sub)section and just presenting the results in section 6.2.

- Page 15, lines 13-14: add "algorithm" to "analytic Gauss-Newton".

- Page 17, lines 7-14: this paragraph is closely related to the paragraph on page 16, lines 16-29. I suggest moving it accordingly.

- Page 17, lines 8-10: please rephrase this sentence to clarify. Do you mean "...exhibit significant positive correlations with Fyw but also statistically negative correlations with the discharge sensitivity of Fyw."?

*Summary and Conclusions

- Here or in previous section: please discuss in a bit more detail the additional information content of the discharge sensitivity. Long-term isotope data of good resolution such as in this study are not a given, so it might be good to know if (what) Fyw can tell us more than "traditional" hydrologic indices addressing flow variability (e.g., CVQ)?

- Page 18, line 31: suggest dropping "however" as this might be confusing to the reader

- Page 19, line 9: suggest replacing "found" by, for example, "hypothesize" as this follows from the conceptual model.

*Figures

- Figure 6: it might be the pdf version, but I can barely discern light blue points.

*Supplement

- suggest adding a map showing the 22 catchments and the 19 long-term monitoring stations for d18O-values in precipitation so the reader can get an idea of the spatial coverage of the measurements. Alternatively, the station locations can be added to Fig. 2.

---

## Referee Comment (RC2) · M. Hrachowitz (Referee) · 10 Jan 2018

This manuscript analysis stable water isotope signals in a range of contrasting catchments in the Swiss Alps to better understand what controls catchment storage and release dynamics. Based on a recently developed metric, the young water fraction (Fyw), the analysis provides a highly interesting and new perspective on the topic: the sensitivity of Fyw to stream flow. From my point of view, this topic alone would already merit publication. In fact, I would even argue that much of the additional analysis provided in the manuscript, specifically the comparison of the interpolation methods and the snow storage considerations, do not really add much value and actually somewhat

dilute the really interesting story. I thus think these parts could easily be removed or at least be considerably shortened, but I leave this decision open to the authors.

Notwithstanding the well-designed experiments and in-depth analysis, the manuscript would benefit from some restructuring and, in places, from more precise and detailed explanations (see detailed comments below). My only major comment is the rather superficial discussion of the relationships between young water fractions and catchment characteristics (section 5). There were quite a lot of studies over the last 10-15 years (e.g. that looked into the relationships of the very same variables, e.g. soil types, L/G, drainage densities, area, TWI, precipitation intensity, etc. , with mean transit times (e.g. McGlynn et al., 2003, HP; McGuire et al., 2005, WRR; Laudon et al., 2007, JoH; Broxton et al., 2009, WRR; Tetzlaff et al., 2009, HP; Hrachowitz et al., 2010, WRR, 2010, HP; Soulsby et al., 2010, HP; Speed et al., 2010, HP; Asano and Uchida, 2012, WRR; Hale and McDonnell, 2016, WRR; and many others). Although the Fyw is an arguably more stable and thus reliable metric, it would be interesting to see and understand how the results and interpretations of the analysis presented here compares to these earlier studies. Can similar conclusion be drawn for Fyw than previously for MTTs? If yes, what does that mean? If no, why? Such a more detailed discussion would lend an additional, interesting edge to the manuscript.

In any case, I would be glad to see this work eventually published and I hope that the authors find my comments helpful.

Detailed comments: (1)P.2, l.8-9: "usually" is a quite unfortunate term here. Clearly, while there are quite some studies using "lumped-parameter" models (I suppose the authors referred to convolution integral approaches), there many(!) other studies that go far beyond that with many different types of models ranging from fully coupled 3D models to more conceptual models based on suites of storage tanks and the associated mixing coefficients/SAS functions. Please rephrase.

(2)P.2, l.10: "catchment storage" is inaccurate. It rather expresses some (essentially

unknown) storage that is significantly affected by exchange processes. For many systems, there may well be significant additional storage below that, which remains essentially undetectable with stable isotope data due to potentially very long time scales of these exchange processes at depth (mostly molecular diffusion?). Please rephrase.

(3)P.2, l.12-13: is this generally true or is it not mostly due to the assumption of time-invariance? Again, please note that most model approaches, except lumped-parameter convolution integral approaches, do *not* rely on time-invariance of TTDs.

(4)P.2,l.16: perhaps better to use "estimated" than "obtained"

(5)P.2,l.13: to be precise, it should read as:". . .from the differences in the amplitudes. . ."

(6)P.3,l.3-21: this is quite lengthy and written in an unnecessarily complicated way. The bottom line is, in my opinion, if only liquid water input to/storage in the system is considered or the total water input/storage.

(7)P.3,l.26: please clarify what is meant by "coefficients" of the seasonal cycles.

(8)P.4,l.5-17: some of the above references, analysing the relationships of catchment characteristics with MTTs would fit in nicely here and would place your manuscript into a somewhat wider context.

(9)P.4,l.32ff: also here, sine-wave fitting has been used already quite long time ago to understand transit times. Please add some references (e.g. DeWalle et al., 1997, HP; Soulsby et al., 2006, JoH)

(10)P.5,l.13: see comment (3)

(11)P.5,l.17ff, eqs.(3) and (4): redundant with eqs.(1) and (2). Instead of amplitude and phase eqs.(3) and (4) give the same information only expressed in sine and cosine components. I think eqs. (1) and (2) can be removed.

(12)P.6,l.26: what does "i. Br." mean?

[Figure]

(13)P.6,l.28: "accuracy" or "precision"?

(14)P.7, section 3.3: also here, some references to earlier papers that used similar and partly the same predictor variables would be good

(15)P.7,l.32: do flow path length and gradient refer to subsurface or total length and gradient to the outlet? Please be more specific.

(16)P.8,l.20-22: was the use of multiple linear regressions considered to better identify potentially spurious correlations? If not, why?

(17)P.8,l.25ff, section 4.1: does this section actually add value to the manuscript? I think, the section can at least be considerably shortened if not condensed altogether.

(18)P.8,l.25-P.9,l.14: this would fit much better into the methods section

(19)P.9,l.28-29: although this term is widely used in our community, I do not think that in any environmental system application we can actually "validate" a model in the actual sense of the word. The best we can do is to rigorously test our models.

(20)P.10,l.1ff, section 4.2: see comment (17). If you decide to keep the section, more detailed descriptions of the model used for the snow dynamics (including parameters, calibration procedure, uncertainties involved, etc.) is needed and can be placed in the supplementary material. In addition, I may have missed it, but it is unclear what PREVAH stands for.

(21)P.11,l.1: not clear what is meant by "...shifts the seasonal isotope pattern toward later in the season." Does this refer to the amplitudes? If yes, please say so.

(22)P.11,l.2-3,fig.4: it would be easier for the reader to appreciate the information content of figure 4, if the phase would be given in days (or months) rather than in radians.

(23)P.11,l.19-23: "...young water fractions...that are larger...because high flows generally contain more young water...". This seems a bit of circular reasoning to me.

(24)P.12,l.3-4: repetition of what was said earlier. Can be omitted.

(25)P.12,l.1ff,section 5: again please see comment (8)

(26)P.12,l.31-34: sure, a few studies could identify area as potential control on MTTs, but others clearly could not (see in the given references above). Thus please rephrase this statement.

(27)P.13,l.28: this interpretation is of course possible, but it surprisingly seems to not consider the potentially important influence of fast, lateral preferential flow pathways (e.g. macropores), which can be abundant in particular at (steep) forest sites. It may be worth reflecting on this a bit more.

(28)P.14,l.4: what is meant by "bigger" cycles?

(29)P.14,l.17: the description of how this was in detail done remains quite vague. Please provide a more detailed description in the methods section. Were samples from time periods outside the individual quartiles simply removed and the sine wave refitted on the remaining samples? How many samples on average were the individual fits then based on? The information content of the 4th quartile and the top 20% is very similar. One can be removed.

(30)P.15,l.1-12: it is not entirely clear in how this is different to what was done in 6.1. Please also here, provide a more detailed description in the methods section of what was done and how.

(31)P.17,l.23-24: which, in turn, would imply (to maintain the fraction of young water in spite of increasingly more young water in the system) an increasingly preferential sampling of older water as the system gets wetter.

(32)P.17,l.26ff: this is a very interesting analysis, but it remains unclear, which parts of it are actually supported by the available data/results and which are mere speculation. Please try to make it clearer, which evidence supports these interpretations.

Best regards, Markus Hrachowitz
* * *
Interactive
comment

---

## Short Comment (SC1) · 25 Jan 2018

**Comments on "Sensitivity of young water fractions to hydro-climatic forcing and landscape properties across 22 Swiss catchments"**

This is a very interesting manuscript about the sensitivity of the young water fraction (Fyw) in streams to discharge and watershed characteristics across many Swiss sites. Although I did not read it as carefully as a reviewer might, it seemed well-reasoned and concluded with an insightful conceptual model informed by novel ideas and analytic techniques.

I am writing to observe that aspects of the paper (referred to hereafter to by the authors' initials FASWK) seem relevant to previous work, including work I co-authored with Harman and Ball and reported in the paper Sensitivity of Catchment Transit Times Under Present and Future Climate (Wilusz et al. 2017, referred to hereafter as WHB). WHB analyzed the relationship between the young water fraction and rainfall variability in 2 subcatchments of the Plynlimon experimental site using a lumped parameter transit time model calibrated to a 10 year data record. I was excited to see that many of the findings in WHB were consistent and complementary to findings in FASWK. I list these points of mutual relevance and complementarity below, in case the authors may also find some of the connections interesting and/or sufficiently relevant to reference in the manuscript.

--- WHB found that every 1mm/day increase in average annual precipitation was associated with a 0.03 and 0.04 increase in the Fyw (WHB, Figure 5d) in the 2 subcatchments studied. In the parlance of FASWK, this metric could be referred to as a kind of "precipitation sensitivity of Fyw". Given the high runoff ratios in the Plynlimon catchments (0.78-0.90, see WHB, Table 2), the precipitation sensitivity of Fyw should be closely related to the discharge sensitivity of Fyw. The values of the precipitation sensitivity of Fyw at Plynlimon multiplied by the runoff-ratio are near the middle of the range of discharge sensitivity of Fyw values reported in the 22 Swiss catchments (FASWK, page 16, line 7-8). The fact that the ranges overlap in the two manuscripts at different (albeit hydrologically similar) sites - even though the models and timescales used for estimation were different – is further evidence that the sensitivity of Fyw to hydro-metric fluxes is a robust and reproducible metric that could "contribute to future (inter-comparison) studies" (FASWK, page 19, line 30) and "be a potentially useful hydrologic signature" (WHB, page 19). Of note, a significant strength of the method proposed in FASWK is that it used lower temporal resolution tracer data, which is more commonly available.

--- WHB found that the annual flow-weighted average Fyw is highly linearly correlated with annual precipitation (WHB, Figure 5d) across time.   This is consistent with the finding in FASWK of a significant linear relationship between Fyw and Pbar (FASWK, Figure 6 upper right panel) across space.

--- The conceptual model in FASWK classifies Case 1 and 3 catchments as having a "constant mixing fraction of young and old water" and Case 2 catchments as where "the relative contribution of fast and slow flowpaths vary dramatic in response to hydro-climatic forcing and antecedent wetness " (FASWK, page 17).  The paper Kim et al. (2016) introduced a related classification scheme, in which the classification "external variability only" was akin to Case 1/3 catchments, and the classification "both internal and external variability" was akin to Case 2 catchments (see Kim et al. 2016, Figure 6).  Kim at al (2016) showed how these two classifications could be mathematically embodied and parameterized in a forward modeling framework using the theory of StorAge Selection (SAS) functions (Botter et al. 2011, van der Velde et al. 2012, Harman 2015).  In addition, analysis in Harman (2015), Kim et al. (2016), Benettin (2017), and WHB showed how a hydrologic system could be analyzed to rigorously test whether it exhibited external only variability (Case 1/3) or external and internal variability (Case 2). (Note a subtle difference between the two classification schemes is that FASWK is based on a distinction between flow pathways that are slow versus fast (as described in Figure 10), while the classification of Kim et al (2016) is based on a distinction between pathways that contribute older age-ranged storage to discharge versus pathways that contribute younger age-ranked storage to discharge.  The difference may be relatively unimportant for the kind of analysis done in FASWK that looks at long-term average behavior in humid catchments.)   To summarize, the relevance of this literature to the FASWK manuscript is: (a) the SAS mathematical framework has been used to rigorously classify watersheds as something similar to Case 1/3 or Case 2; (b) the parameterization of SAS functions could be informed by its designation as either Case 1, 2 or 3; and (c) the parameterization of SAS functions could be informed by the relationships reported in FASWK between the Fyw and watershed properties.

--- WHB incorporated the sensitivity of the Fyw to hydro-climatic forcing into a forward modelling framework to do a first-order projection of the impact of climate change on the Fyw at the Plynlimon sites.  WHB projections showed the Fyw would decrease significantly in summer, and increase significantly in winter.  This illustrates one of many ways information about the the sensitivity of Fyw to hydro-climatic forcing could be used to help answer management relevant questions, as suggested in FASWK page 18, lines 14-17.

--- The use of the sensitivity of the Fyw to hydro-climatic forcing and landscape properties for intercatchment comparison behavior has roots in the literature.  For example, Harman (2015) defined and proposed using a "sensitivity of event water fraction to discharge" (Harman 2015, page 23) as a useful transport-sensitivity metric.  As discussed above, WHB used something akin to a "precipitation sensitivity of Fyw" for a 2-catchment comparison. WHB also has a brief literature review summarizing previous work relating age distributions to hydro-climatic fluxes (WHB, section 1.1).  In addition, some researchers are using SAS functions for catchment classification and intercomparison (see for example Rinaldo et al. 2015), and SAS functions could be seen as a generalization of the discharge sensitivity of Fyw, to the extent that knowledge of SAS functions and flux history is sufficient to estimate the discharge sensitivity of Fyw for any control volume of interest.

Dano Wilusz

PhD Candidate, Johns Hopkins University

**References**

Botter, G., Bertuzzo, E., & Rinaldo, A. (2011). Catchment residence and travel time distributions: The master equation. Geophysical Research Letters, 38, L11403. https://doi.org/10.1029/2011GL047666

Benettin, P., Soulsby, C., Birkel, C., Tetzlaff, D., Botter, G., & Rinaldo, A. (2017). Using SAS functions and high-resolution isotope data to unravel travel time distributions in headwater catchments. Water Resources Research, 53(3), 1864-1878.

Harman, C. J. (2015). Time-variable transit time distributions and transport: Theory and application to storage-dependent transport of chloride in a watershed. Water Resources Research, 51(1), 1-30.

Kim, M., Pangle, L. A., Cardoso, C., Lora, M., Volkmann, T. H., Wang, Y., ... & Troch, P. A. (2016). Transit time distributions and StorAge Selection functions in a sloping soil lysimeter with time-varying flow paths: Direct observation of internal and external transport variability. Water Resources Research, 52(9), 7105-7129.

Rinaldo, A., Benettin, P., Harman, C. J., Hrachowitz, M., McGuire, K. J., Van Der Velde, Y., ... & Botter, G. (2015). Storage selection functions: A coherent framework for quantifying how catchments store and release water and solutes. Water Resources Research, 51(6), 4840-4847.

van der Velde, Y., Torfs, P. J. J. F., van der Zee, S. E. A. T. M., & Uijlenhoet, R. (2012). Quantifying catchment-scale mixing and its effect on time-varying travel time distributions. Water Resources Research, 48, W06536. https://doi.org/10.1029/2011WR011310

Wilusz, D. C., Harman, C. J., & Ball, W. P. Sensitivity of Catchment Transit Times to Rainfall Variability Under Present and Future Climates. Water Resources Research, 53.

---

## Author Response (AR1)

Zurich, 25 April 2018

Dear Dr. Rolf Merz,

Please find attached the revised version of our manuscript entitled "Sensitivity of young water fractions to hydro-climatic forcing and landscape properties across 22 Swiss catchments" (hess-2017-720). We addressed all issues raised by the reviewers and you can find our detailed responses, the track-changed manuscript, as well as the track-changed Supplementary Material below.

In the introduction, we have moved the section about the representation of the snowpack as part of the catchment storage further down in order to be consistent with the order of Sect. 4. Further, following both reviewers' suggestions, we have compressed section 4.1 about the two precipitation isotope interpolation methods by moving the methods description into the Methods section (new: Sect. 3.4). With this, Section 4 becomes considerably shorter.

Method 2, which was used in our study for interpolating precipitation isotope values and which is described in detail in the Supplement, is based on the approach developed by one of our Co-authors, Scott T. Allen. Unfortunately, the manuscript of Allen et al. is still under review so that we could not update the reference in our manuscript accordingly. Thus, we cite this study as "Allen et al. (submitted manuscript)".

We highly appreciated the thoughtful comments of you and the two reviewers and the short comment, which helped to improve the manuscript.

Thank you very much,

Jana von Freyberg et al.

**Response to the interactive comment of Reviewer #1 on**

"Sensitivity of young water fractions to hydro-climatic forcing and landscape properties across 22 Swiss catchments" by Jana von Freyberg et al.

**General comments**

*This discussion paper presents an analysis of young water fractions (Fyw) in contrasting catchments across Switzerland. The paper first examines the influence of interpolation methods, flow-weighting of measurements and snow in the calculation of Fyw. The second part studies correlations between young water fractions and catchment characteristics. The authors then introduce a new metric (i.e., the discharge sensitivity of Fyw) and relate this metric to catchment characteristics. The paper concludes with a conceptualization of the relationship between young water fractions and streamflow. The paper is well written and addresses important problems in the analysis of isotope data, i.e., interpolation, impact of snow and flow-weighting. Moreover, the paper presents a new concept derived from the recently introduced young water fractions. However, there are some parts that need clarification and rearranging especially in the theoretical and methodological sections.*

We thank Reviewer #1 for his/her thoughtful comments, which helped to improve the manuscript. Please find our detailed responses below.

*Comments of the reviewer are shown in italics.* Responses from the authors are presented in regular font below each comment.  Citations from the manuscript are in Times New Roman, changes of the cited manuscript text are underlined.

**Specific comments**

**\*Abstract**

*- Page 1, line 23: suggest clarifying what "this relationship" is (i.e., the relationship between flow and young water fractions).*

We will change that.

**\*Theoretical background**

*- Page 5, lines 17-20: please clarify that equations (3) and (4) follow from (1) and (2).*

We will change that.

*- Page 5, lines 25-27: it is not entirely clear what is meant with volume-weighting. I assume it does not refer to the isotope values themselves (so volume-weighting over several samples to obtain a weighted catchment average, as done for the precipitation isotope values), but to the weighting scheme within the IRLS algorithm.*

Within the iteratively reweighted least-squares (IRLS) algorithm we allow for optional point weights (in addition to residual weights that are adjusted to down-weight data with unusually large residuals, as in conventional IRLS).  In the R-script provided in the supplementary material the user can choose between three weighting functions: Bi-square, Welsh or Cauchy.  In our analyses, the weighting of the isotope data was carried out with the Cauchy weight function.

**\*Data set**

*- Page 8, lines 6-10: suggest dropping the German terms of the soil properties as this will not mean anything to most readers.*

We will change that.

**Results/Discussion*

*- Due to the concise description of the interpolation methods in the main next, it is not easy for the reader to follow the different steps of the two interpolation methods, although this would be helpful to better understand the differences between the two methods. Moreover, method 2 has been developed by the authors, so this method should be introduced more extensively in the main text. I would thus suggest restructuring the paper by moving major parts of the methodology description from the Supplement to the main text. This could be placed into a subsection of section 3 or a separate methodological section. Please also explain method 2 in a bit more detail – in the main text, this method is described with one long sentence only. The comparison between the two methods can be kept in section 4.1, which would be more consistent with presenting results only in section 4.*

We will move this part into Chapter 3 and keep the short discussion of the results in Section 4.1. With this, Chapter 4 becomes considerably shorter, while both interpolation methods are still described in a short manner in the manuscript (new: Sect. 3.4 Precipitation isotope data). A detailed description of method 2 will still be available in the Supplement.

*- Page 8, line 26: are these cumulative monthly d18O-values in precipitation (so sampling bottle emptied each month)?*

Yes, the GNIP reports isotope values from cumulative samples. We will clarify this.

*- Page 9, line 25; page 11, line 5 and line 16: "statistically (in)significant" using which statistical method?*

We will include a definition of that term in Page 9, line 25: "… (i.e., smaller than twice their pooled uncertainties, Figure 3b)."

*- Page 11, line 25: this is the first time the authors mention "gamma distributions". Please clarify that this refers to the underlying transit time distribution model.*

We will clarify that: "The average values of $F_{yw}^*$ and $F_{yw}$ were 0.22±0.02 and 0.17±0.02, respectively, meaning that approximately 1/5 of total discharge was younger than roughly 2.3±0.8 months (assuming that the catchment transit times can be described by gamma distributions with shape factors $\alpha$ ranging from 0.3 to 2)."

*- Page 12, lines 29-31: suggest weakening this statement ("consistent with . . .") as results from a global analysis should be compared with caution to regional analyses and the smaller Fyw in this study could also be caused by various factors other than the gradient dependence. See also page 19, lines 7-8.*

With this comparison we aim to put our regional results into a global context. However, we do acknowledge that the range of young water fractions is wide ("[…] 10th to 80th percentiles of the $F_{yw}$ values estimated by Jasechko et al. (2016) […]"), which suggests that other factors than gradient are likely controlling the discharge of young water. This is further analyzed in the following section 5

"Relationships between young water fractions, hydro-climatic conditions and landscape characteristics"

*- Page 14, lines 15-20: please give a bit more details on the procedure: how many measurements were on average available in each sine-wave regression after separation by flow regimes? Was the number of values sufficient to obtain reliable results? I would expect the seasonal variations to be small and potentially indiscernible under low-flow conditions, when streamflow is dominated by the well-mixed signal of slow flowpaths.*

The separation of the flow regimes was carried out in dependence of the flow at the time of sampling, so that roughly similar numbers of data points were available for each flow regime. For instance, at the Erlenbach site, the total number of streamwater isotope samples was 140, and thus each quartile of $Q$ comprised 35 samples, while the upper 20 % and 10 %, of daily discharges comprised 28 and 14 samples, respectively. At other sites with much smaller numbers of streamwater samples, this separation procedure would not yield enough isotope samples to reliably estimate $F_{yw}$ for each flow regime. Therefore, we used the alternative approaches presented in the following Sect. 6.2.

*- Page 15, line 3 – page 16, line 4: suggest introducing the concept of discharge sensitivity earlier in the manuscript as a methodological (sub)section and just presenting the results in section 6.2.*

We would like to keep the current order of the manuscript as it would possibly cause confusion to present the discharge sensitivity too early in the manuscript (i.e., in Sects. 2 or 3) before the strong linkages between catchment wetness and young water fractions could be established. The discharge sensitivity analysis in Sect. 6 consequences immediately from the comparison of flow-weighted versus unweighted young water fractions (Sect. 4.3) and the catchment-comparison analysis (Sect. 5).

*- Page 15, lines 13-14: add "algorithm" to "analytic Gauss-Newton".*

We will change that.

*- Page 17, lines 7-14: this paragraph is closely related to the paragraph on page 16, lines 16-29. I suggest moving it accordingly.*

We agree with the reviewer that Page 17, lines 7-14 repeats some of the results presented previously, however, we would like to keep this paragraph as it is to better compare the opposite correlations of the young water fraction and its discharge sensitivity with respect to the catchment characteristics.

*- Page 17, lines 8-10: please rephrase this sentence to clarify. Do you mean ". . .exhibit significant positive correlations with Fyw but also statistically negative correlations with the discharge sensitivity of Fyw."?*

We will change that.

*Summary and Conclusions*

*- Here or in previous section: please discuss in a bit more detail the additional information content of the discharge sensitivity. Long-term isotope data of good resolution such as in this study are not a given, so it might be good to know if (what) Fyw can tell us more than "traditional" hydrologic indices addressing flow variability (e.g., CVQ)?*

Traditional hydrometric metrics such as *CVQ* of *QFI* solely allow to draw conclusions about the response times of a catchment, while no information can be obtained about how much young water a flood peak contains. In contrast, the discharge sensitivity expresses how the fraction of young water changes with catchment wetness (expressed by *Q*), and thus we gain more information about the storage behavior of a catchment.

*- Page 18, line 31: suggest dropping "however" as this might be confusing to the reader*

We will change that.

*- Page 19, line 19: suggest replacing "found" by, for example, "hypothesize" as this*

*follows from the conceptual model.*

We will change that.

***Figures***

*- Figure 6: it might be the pdf version, but I can barely discern light blue points.*

We will increase the color contrast between the data points shown in Fig. 6.

***Supplement***

*- suggest adding a map showing the 22 catchments and the 19 long-term monitoring stations for d18O-values in precipitation so the reader can get an idea of the spatial coverage of the measurements. Alternatively, the station locations can be added to Fig. 2.*

We will include an overview map of the stations in the Supplement.

"Sensitivity of young water fractions to hydro-climatic forcing and landscape properties across 22 Swiss catchments" by Jana von Freyberg et al.

*This manuscript analysis stable water isotope signals in a range of contrasting catchments in the Swiss Alps to better understand what controls catchment storage and release dynamics. Based on a recently developed metric, the young water fraction (Fyw), the analysis provides a highly interesting and new perspective on the topic: the sensitivity of Fyw to stream flow. From my point of view, this topic alone would already merit publication. In fact, I would even argue that much of the additional analysis provided in the manuscript, specifically the comparison of the interpolation methods and the snow storage considerations, do not really add much value and actually somewhat dilute the really interesting story. I thus think these parts could easily be removed or at least be considerably shortened, but I leave this decision open to the authors. Notwithstanding the well-designed experiments and in-depth analysis, the manuscript would benefit from some restructuring and, in places, from more precise and detailed explanations (see detailed comments below).*

*My only major comment is the rather superficial discussion of the relationships between young water fractions and catchment characteristics (section 5). There were quite a lot of studies over the last 10-15 years (e.g. that looked into the relationships of the very same variables, e.g. soil types, L/G, drainage densities, area, TWI, precipitation intensity, etc., with mean transit times (e.g. McGlynn et al., 2003, HP; McGuire et al., 2005, WRR; Laudon et al., 2007, JoH; Broxton et al., 2009, WRR; Tetzlaff et al., 2009, HP; Hrachowitz et al., 2010, WRR, 2010, HP; Soulsby et al., 2010, HP; Speed et al., 2010, HP; Asano and Uchida, 2012, WRR; Hale and McDonnell, 2016, WRR; and many others). Although the Fyw is an arguably more stable and thus reliable metric, it would be interesting to see and understand how the results and interpretations of the analysis presented here compares to these earlier studies. Can similar conclusion be drawn for Fyw than previously for MTTs? If yes, what does that mean? If no, why? Such a more detailed discussion would lend an additional, interesting edge to the manuscript. In any case, I would be glad to see this work eventually published and I hope that the authors find my comments helpful.*

We thank Dr. Hrachowitz for his thoughtful comments, which we have addressed in detail below.

*Comments of the reviewer are shown in italics.* Responses from the authors are presented in regular font below each comment. Citations from the manuscript are in Times New Roman, changes of the manuscript text are underlined.

**Detailed comments:**

*(1)P.2, l.8-9: "usually" is a quite unfortunate term here. Clearly, while there are quite some studies using "lumped-parameter" models (I suppose the authors referred to convolution integral approaches), there many(!) other studies that go far beyond that with many different types of models ranging from fully coupled 3D models to more conceptual models based on suites of storage tanks and the associated mixing coefficients/SAS functions. Please rephrase.*

*We will change that:* "Transit time distributions are often inferred from concentrations of conservative tracers, such as stable water isotopes in precipitation and streamwater, using lumped-parameter models…"

*(2)P.2, l.10: "catchment storage" is inaccurate. It rather expresses some (essentially unknown) storage that is significantly affected by exchange processes. For many systems, there may well be significant additional storage below that, which remains essentially undetectable with stable isotope*

*data due to potentially very long time scales of these exchange processes at depth (mostly molecular diffusion?). Please rephrase.*

We will clarify this by writing: "Because the mean transit time expresses the ratio between mobile catchment storage and the average flow rate, it is widely used in catchment inter-comparison studies…"

*(3)P.2, l.12-13: is this generally true or is it not mostly due to the assumption of time-invariance? Again, please note that most model approaches, except lumped parameter convolution integral approaches, do \*not\* rely on time-invariance of TTDs.*

The cited references substantiate the problem of bias and unreliability in estimates of mean transit times. The conjecture that this problem disappears when TTD's are assumed to be time-variant is interesting but, as far as we know, not (yet) demonstrated – and it side-steps the very difficult problem of estimating what time-varying transit time distributions actually are (in the real world, based on real-world data, not in models).

*(4)P.2,l.16: perhaps better to use "estimated" than "obtained"*

We will change that.

*(5)P.2,l.13: to be precise, it should read as:". . .from the differences in the amplitudes. . ."*

Strictly speaking, it is the ratio of the seasonal cycle amplitudes that defines the young water fraction. We will clarify this in the revised version of the manuscript.

*(6)P.3,l.3-21: this is quite lengthy and written in an unnecessarily complicated way. The bottom line is, in my opinion, if only liquid water input to/storage in the system is considered or the total water input/storage.*

We find it important to properly explain both cases (catchment storage including/excluding snow storage) to the reader so that the relevance of this distinction regarding the interpretation of young water fractions can be grasped. Since the young water fraction framework is/will be applied to catchments in very different climatic regimes, we would like to emphasize these conceptual descriptions of catchment storage early on in the manuscript.

*(7)P.3,l.26: please clarify what is meant by "coefficients" of the seasonal cycles.*

We will write "seasonal cycle amplitudes", instead.

*(8)P.4,l.5-17: some of the above references, analysing the relationships of catchment characteristics with MTTs would fit in nicely here and would place your manuscript into a somewhat wider context.*

We will include some of the references in the Introduction: "Because the young water fraction can be estimated from sparse and irregular tracer data, it has been suggested as a useful metric for catchment inter-comparison studies (Kirchner, 2016). To date, however, most catchment inter-comparison studies have investigated controls on mean transit times instead. Mean transit times have been variably found to be correlated with (for example) flow path lengths and gradients (McGuire et al., 2005), drainage density (Soulsby et al., 2010), the areal fraction of hydrologically responsive soils (Tetzlaff et al., 2009), bedrock permeability (Hale and McDonnell, 2016), or combinations of multiple factors (Hrachowitz et al., 2009; Seeger and Weiler, 2014).

*(9)P.4,l.32ff: also here, sine-wave fitting has been used already quite long time ago to understand transit times. Please add some references (e.g. DeWalle et al., 1997, HP; Soulsby et al., 2006, JoH)*

We will include these references.

*(10)P.5,l.13: see comment (3)*

Please, refer to our reply to comment (3).

*(11)P.5,l.17ff, eqs.(3) and (4): redundant with eqs.(1) and (2). Instead of amplitude and phase eqs.(3) and (4) give the same information only expressed in sine and cosine components. I think eqs. (1) and (2) can be removed.*

We would like to keep Eqs.(1) and (2) as they introduce the general idea of fitting sine curves with coefficients $A$ and $k$ to streamwater or precipitation isotope time series. By only presenting Eqs. (3) and (4) instead, the meaning of $A$ and $k$ might not be clear without explanation. We thus consider Eqs. (1) and (2) the most efficient way to do this.

*(12)P.6,l.26: what does "i. Br." mean?*

It means "im Breisgau". We have removed this expression in the main text but kept it in the affiliations of the authors.

*(13)P.6,l.28: "accuracy" or "precision"?*

Accuracy, which is consistent with the information given in Seeger et al. (2014).

*(14)P.7, section 3.3: also here, some references to earlier papers that used similar and partly the same predictor variables would be good*

We will add more references here.

*(15)P.7,l.32: do flow path length and gradient refer to subsurface or total length and gradient to the outlet? Please be more specific.*

We used the same indices as in Seeger et al. (2014), which were derived with the SAGA module "Overland Flow Distance to Channel network". The flow path length $L$ refers to the total (surface) length of the stream network, while the gradient $G$ was calculated from the ratio of the horizontal and vertical components of L ($L_h$ and $L_v$). We will clarify this in the revised manuscript.

*(16)P.8,l.20-22: was the use of multiple linear regressions considered to better identify potentially spurious correlations? If not, why?*

We tried this. Multiple linear regression analysis on such a small sample (22 sites), with such a large number of candidate explanatory variables, leads to results that are strongly dependent on the specific model selection criteria that are used. Thus, we feel that a simple table of rank correlations is a more realistic, if more modest, representation of our results.

*(17)P.8,l.25ff, section 4.1: does this section actually add value to the manuscript? I think, the section can at least be considerably shortened if not condensed altogether.*

*(18)P.8,l.25-P.9,l.14: this would fit much better into the methods section*

In contrast to Reviewer #2, Reviewer #1 asked to expand more on the description of methods 1 and 2, and therefore we will move this part into Chapter 3 and keep the short discussion of the results in Section 4.1.  With this, Chapter 4 becomes considerably shorter, while both interpolation methods are still described in a short manner in the manuscript (new: Sect. 3.4 Precipitation isotope data).  A detailed description of method 2 will still be available in the Supplement.

*(19)P.9,l.28-29: although this term is widely used in our community, I do not think that in any environmental system application we can actually "validate" a model in the actual sense of the word. The best we can do is to rigorously test our models.*

We will change that.

*(20)P.10,l.1ff, section 4.2: see comment (17). If you decide to keep the section, more detailed descriptions of the model used for the snow dynamics (including parameters, calibration procedure, uncertainties involved, etc.) is needed and can be placed in the supplementary material. In addition, I may have missed it, but it is unclear what PREVAH stands for.*

PREVAH stands for PREcipitation-Runoff-EVApotranspiration HRU model.  It was used here to interpolate hydro-climatic variables at the study sites (See Sect. 3.1 Hydro-climatic data).

*(21)P.11,l.1: not clear what is meant by ". . .shifts the seasonal isotope pattern toward later in the season." Does this refer to the amplitudes? If yes, please say so.*

We will change that: "As can be seen in Figure 4a, the delayed meltwater input shifts the phase of the seasonal isotope pattern toward later in the season."

*(22)P.11,l.2-3,fig.4: it would be easier for the reader to appreciate the information content of figure 4, if the phase would be given in days (or months) rather than in radians.*

We will change Fig. 4 to show the phase and phase shifts in fractions of 1 year instead of radians.

*(23)P.11,l.19-23: "...young water fractions. . .that are larger. . .because high flows generally contain more young water. . .". This seems a bit of circular reasoning to me.*

We argue that the statement "…because high flows generally contain more young water..." puts the increase in young water fractions after flow-weighting into a process-based context.  We therefore won't change the sentence.

*(24)P.12,l.3-4: repetition of what was said earlier. Can be omitted.*

We will remove this sentence.

*(25)P.12,l.1ff,section 5: again please see comment (8)*

We agree with the reviewer, that numerous earlier studies have looked into the relationship between MTT's and catchment characteristics. However, for the reasons outlined in the Introduction of the manuscript (an in much more detail by Kirchner (2016a,b)), that is that MTT's are prone to severe aggregation bias and are thus likely to be uncertain, we don't think a direct comparison of our results with those earlier studies is useful.

*(26)P.12,l.31-34: sure, a few studies could identify area as potential control on MTTs, but others clearly could not (see in the given references above). Thus please rephrase this statement.*

We did not claim that this is a universal relationship, but rather indicate that some studies did find a significant correlation. We will change the sentence, to make this more clear: "Some studies have identified catchment area as a major control on mean transit times (e.g., DeWalle et al., 1997; Soulsby et al., 2000), however, the negative correlation of $F_{yw}^*$ and $F_{yw}$ with catchment area only becomes significant ($\rho$=-0.49, $p$<0.05) when the five high-elevation, snow-dominated sites are omitted from the analysis (Fig. 6)"

*(27)P.13,l.28: this interpretation is of course possible, but it surprisingly seems to not consider the potentially important influence of fast, lateral preferential flow pathways (e.g. macropores), which can be abundant in particular at (steep) forest sites. It may be worth reflecting on this a bit more.*

We will add this alternative explanation to the revised version of the manuscript: "One would normally expect tree roots to increase soil permeability, resulting in greater infiltration and groundwater recharge (Brantley et al., 2017). However, at steep, forested slopes, abundant lateral preferential flow pathways (e.g. macropores) may facilitate rapid transport of water (Whipkey, 1965)."

*(28)P.14,l.4: what is meant by "bigger" cycles?*

With bigger we mean that the seasonal cycles of streamwater isotopes are less dampened, which is redundant with larger values of Fyw. We will remove this part of the sentence.

*(29)P.14,l.17: the description of how this was in detail done remains quite vague. Please provide a more detailed description in the methods section. Were samples from time periods outside the individual quartiles simply removed and the sine wave refitted on the remaining samples? How many samples on average were the individual fits then based on? The information content of the 4th quartile and the top 20% is very similar. One can be removed.*

The separation of the flow regimes was carried out in dependence of the flow at the time of sampling, so that roughly similar numbers of data points were available for each flow regime. For instance, at the Erlenbach site, the total number of streamwater isotope samples was 140, and thus each quartile of $Q$ comprised 35 samples, while the upper 20 % and 10 %, of daily discharges comprised 28 and 14 samples, respectively. At other sites with much smaller numbers of streamwater samples, this separation procedure would not yield enough isotope samples to reliably estimate $F_{yw}$ for each flow regime. Therefore, we used the alternative approaches presented in section 6.2. We will include a more detailed description of this approach in the revised version of the manuscript.

*(30)P.15,l.1-12: it is not entirely clear in how this is different to what was done in 6.1. Please also here, provide a more detailed description in the methods section of what was done and how.*

We will add an explanatory sentence: *"As a first-order estimate of the sensitivity of $F_{yw}$ to discharge across all 22 study catchments, we calculated the linear slope of the relationship between $Q$ and $F_{yw}$, using a method that does not require breaking the streamwater isotope time series into separate flow regimes (and thus has more modest data requirements than plots like Figure 7). Thus, instead of fitting a linear slope to the few data points shown in Figure 7, we estimated the linear slope of the $Q$-$F_{yw}$ relationship directly from the tracer time series $c_S(t)$ and $c_P(t)$. For each site, we assume that […]"*

*(31)P.17,l.23-24: which, in turn, would imply (to maintain the fraction of young water in spite of increasingly more young water in the system) an increasingly preferential sampling of older water as the system gets wetter.*

This scenario is possible (besides the scenario that the age distribution remains the same with increasing streamflow), however, we can only speculate about this.

*(32)P.17,l.26ff: this is a very interesting analysis, but it remains unclear, which parts of it are actually supported by the available data/results and which are mere speculation. Please try to make it clearer, which evidence supports these interpretations.*

The analysis in Sect. 6.3 is based on the findings presented in Sects. 6.1 and 6.2. We will include these references to make this more clear to the reader. Besides that, we link the interpretations to the results they are based on by referencing specific figures of some study sites for which the three cases are distinguishable.

*This is a very interesting manuscript about the sensitivity of the young water fraction (Fyw) in streams to discharge and watershed characteristics across many Swiss sites. Although I did not read it as carefully as a reviewer might, it seemed well-reasoned and concluded with an insightful conceptual model informed by novel ideas and analytic techniques.*
*I am writing to observe that aspects of the paper (referred to hereafter to by the authors' initials FASWK) seem relevant to previous work, including work I co-authored with Harman and Ball and reported in the paper Sensitivity of Catchment Transit Times Under Present and Future Climate (Wilusz et al. 2017, referred to hereafter as WHB). WHB analyzed the relationship between the young water fraction and rainfall variability in 2 subcatchments of the Plynlimon experimental site using a lumped parameter transit time model calibrated to a 10 year data record. I was excited to see that many of the findings in WHB were consistent and complementary to findings in FASWK. I list these points of mutual relevance and complementarity below, in case the authors may also find some of the connections interesting and/or sufficiently relevant to reference in the manuscript.*

We thank Mr. Wilusz for commenting on our work. We have replied to his remarks below.
*Comments of Mr. Wilusz are shown in italics.* Responses from the authors are presented in regular font below each comment. Citations from the manuscript are in Times New Roman, changes of the cited manuscript text are underlined.

1.  *WHB found that every 1mm/day increase in average annual precipitation was associated with a 0.03 and 0.04 increase in the Fyw (WHB, Figure 5d) in the 2 subcatchments studied. In the parlance of FASWK, this metric could be referred to as a kind of "precipitation sensitivity of Fyw". Given the high runoff ratios in the Plynlimon catchments (0.78-0.90, see WHB, Table 2), the precipitation sensitivity of Fyw should be closely related to the discharge sensitivity of Fyw. The values of the precipitation sensitivity of Fyw at Plynlimon multiplied by the runoff-ratio are near the middle of the range of discharge sensitivity of Fyw values reported in the 22 Swiss catchments (FASWK, page 16, line 7-8). The fact that the ranges overlap in the two manuscripts at different (albeit hydrologically similar) sites - even though the models and timescales used for estimation were different – is further evidence that the sensitivity of Fyw to hydro-metric fluxes is a robust and reproducible metric that could "contribute to future (inter-comparison) studies" (FASWK, page 19, line 30) and "be a potentially useful hydrologic signature" (WHB, page 19). Of note, a significant strength of the method proposed in FASWK is that it used lower temporal resolution tracer data, which is more commonly available.*

We agree with Mr. Wilusz that the overlap in ranges of the discharge sensitivity between the Plynlimon sites and the Swiss catchments is an interesting finding. We will include this comparison in the revised version of the manuscript.

P16, L5: "At the Aach catchment, only two streamwater samples were collected during high-flow conditions, resulting in an unrealistic and highly uncertain value for $m_S$. At the remaining 21 sites, the linear slopes of the $Q$-$F_{yw}$ relationships range between zero (within error) at Ilfis and Sitter, and $0.0732\pm0.0360$ d mm$^{-1}$ at Mentue, with an average value of $0.0202\pm0.0046$ d mm$^{-1}$. On average, we find that every 1mm day$^{-1}$ increase in discharge is associated with an increase of $0.0202\pm0.0046$ in $F_{yw}$. From this analysis, we excluded the Aach catchment because only two streamwater samples were collected during high-flow conditions, resulting in an unrealistic and highly uncertain value for $m_S$. At the remaining 21 sites, the discharge sensitivities of $F_{yw}$ range between zero (within error) at Ilfis and Sitter, and $0.0732\pm0.0360$ d mm$^{-1}$ at Mentue. These values are similar to those found by Wilusz et al. (2017) for two neighbouring catchments in Plynlimon, Wales. For the two sites, Wilusz et al. (2017) combined a rainfall-runoff model with a rank StorAge Selection (rSAS) transit time model and estimated an increase in $F_{yw}$ by 0.031 to 0.040, respectively, with every 1mm day$^{-1}$ increase in average annual precipitation. Multiplying their "precipitation sensitivities of $F_{yw}$" by the site-specific runoff ratios (0.78 and 0.90) yields average discharge sensitivities of $F_{yw}$ of 0.0242 and 0.0360 d mm$^{-1}$, respectively, which are within the range of values we obtained for our 22 Swiss study sites. Even though the methods, tracers and timescales Wilusz et al. used to estimate $F_{yw}$ differed from ours, the similarity in the discharge sensitivities between their sites and ours suggests that this may be a robust and reproducible metric that could be useful in future catchment (inter-comparison) studies."

2. *WHB found that the annual flow-weighted average Fyw is highly linearly correlated with annual precipitation (WHB, Figure 5d) across time. This is consistent with the finding in FASWK of a significant linear relationship between Fyw and Pbar (FASWK, Figure 6 upper right panel) across space.*

One should of course expect $F_{yw}$ to be higher under wetter conditions as a general rule, but our results and those of WHB are apples and oranges. WHB compare model results across years; we show site-to-site comparisons based on real-world data. Naturally WHB's model results show a strong correlation; the internal consistency of model behavior all but guarantees this.

3. *The conceptual model in FASWK classifies Case 1 and 3 catchments as having a "constant mixing fraction of young and old water" and Case 2 catchments as where "the relative contribution of fast and slow flowpaths vary dramatic in response to hydro-climatic forcing and antecedent wetness " (FASWK, page 17). The paper Kim et al. (2016) introduced a related classification scheme, in which the classification "external variability only" was akin to Case 1/3 catchments, and the classification "both internal and external variability" was akin to Case 2 catchments (see Kim et al. 2016, Figure 6). Kim at al (2016) showed how these two classifications could be mathematically embodied and parameterized in a forward modeling framework using the theory of StorAge Selection (SAS) functions (Botter et al. 2011, van der Velde et al. 2012, Harman 2015). In addition, analysis in Harman (2015), Kim et al. (2016), Benettin (2017), and WHB showed how a hydrologic system could be analyzed to rigorously test whether it exhibited external only variability (Case 1/3) or external and internal variability (Case 2). (Note a subtle difference between the two classification schemes is that FASWK is based on a distinction between flow pathways that are slow versus fast (as described in Figure 10), while the classification of Kim et al (2016) is based on a distinction between pathways that contribute older age-ranged storage to discharge versus pathways that contribute younger age-ranked storage to discharge. The*

*difference may be relatively unimportant for the kind of analysis done in FASWK that looks at long-term average behavior in humid catchments.) To summarize, the relevance of this literature to the FASWK manuscript is: (a) the SAS mathematical framework has been used to rigorously classify watersheds as something similar to Case 1/3 or Case 2; (b) the parameterization of SAS functions could be informed by its designation as either Case 1, 2 or 3; and (c) the parameterization of SAS functions could be informed by the relationships reported in FASWK between the Fyw and watershed properties.*

We do not see a clear conceptual link between our classification scheme and that of Kim et al. (2016). It is also not clear how useful SAS functions will be for "rigorously classifying" watersheds, given the apparent difficulty in accurately estimating SAS functions from field data. One of the major advantages of the $F_{yw}$ approach is that it can be applied to catchments where extensive tracer data are not available.

4. *WHB incorporated the sensitivity of the Fyw to hydro-climatic forcing into a forward modelling framework to do a first-order projection of the impact of climate change on the Fyw at the Plynlimon sites. WHB projections showed the Fyw would decrease significantly in summer, and increase significantly in winter. This illustrates one of many ways information about the sensitivity of Fyw to hydro-climatic forcing could be used to help answer management relevant questions, as suggested in FASWK page 18, lines 14-17.*

We thank Mr. Wilusz for this remark, which we will implement into the revised version of the manuscript: "Based on our analysis, we developed a generalized conceptual description that relates $F_{yw}$ and its discharge sensitivity to dominant streamflow generation mechanisms (Sect. 6.3., Fig. 10), which could be useful for analysing the effects of future climate change on catchment hydrologic behavior. It remains to be tested […]"

5. *The use of the sensitivity of the Fyw to hydro-climatic forcing and landscape properties for intercatchment comparison behavior has roots in the literature. For example, Harman (2015) defined and proposed using a "sensitivity of event water fraction to discharge" (Harman 2015, page 23) as a useful transport-sensitivity metric. As discussed above, WHB used something akin to a "precipitation sensitivity of Fyw" for a 2-catchment comparison. WHB also has a brief literature review summarizing previous work relating age distributions to hydro-climatic fluxes (WHB, section 1.1). In addition, some researchers are using SAS functions for catchment classification and intercomparison (see for example Rinaldo et al. 2015), and SAS functions could be seen as a generalization of the discharge sensitivity of Fyw, to the extent that knowledge of SAS functions and flux history is sufficient to estimate the discharge sensitivity of Fyw for any control volume of interest.*

The concept of linking event water fractions (or young water fractions) to hydro-climatic indices is not new (either to our work or that of WHB), and indeed, most of our correlation analysis (Sect. 5 and 6.2) was inspired by those earlier studies (which we reference accordingly). Thus, we do not claim to have invented the expression "discharge sensitivity of $F_{yw}$" as a novel concept of looking at

these relationships. We rather introduce the expression in Sect. 6.2 for reasons of convenience, i.e. instead of using the lengthier expression "linear slope of the $Q$-$F_{yw}$-relationship".

**Table S1: Elevation ranges, as well as hydrologic soil properties and hydrogeological characteristics of the 22 Swiss study catchments.**

| Catchment name | Phase of seasonality of monthly precipitation $\varphi_{precip}$ (months) | Fraction of shallow soils (%) | Fraction of low - medium water storage capacity soils (%) | Fraction of high - very high permeability soils (%) | Fraction of aquifers with low productivity (%) | Fraction of aquifers with intermediate productivity (%) | Fraction of aquifers with high productivity (%) |
|---|---|---|---|---|---|---|---|
| Aabach | 3.8 | 0 | 0 | 23 | 87 | 0 | 13 |
| Aach | 3.9 | 0 | 0 | 0 | 86 | 0 | 14 |
| Allenbach | 3.2 | 78 | 57 | 30 | 89 | 11 | 0 |
| Alp | 3.2 | 68 | 48 | 1 | 81 | 6 | 12 |
| Biber | 3.3 | 30 | 30 | 0 | 94 | 0 | 6 |
| Dischmabach | 3.5 | 59 | 59 | 59 | 92 | 9 | 0 |
| Emme | 3.3 | 78 | 49 | 21 | 88 | 11 | 0 |
| Ergolz | 4.0 | 42 | 41 | 28 | 42 | 54 | 5 |
| Erlenbach | 3.2 | 100 | 4 | 0 | 82 | 18 | 0 |
| Guerbe | 3.6 | 48 | 35 | 45 | 70 | 17 | 13 |
| Ilfis | 3.3 | 32 | 28 | 42 | 92 | 1 | 7 |
| Langeten | 3.4 | 0 | 0 | 37 | 77 | 13 | 10 |
| Lümpenenbach | 3.1 | 100 | 4 | 0 | 100 | 0 | 0 |
| Mentue | 4.7 | 0 | 0 | 76 | 99 | 0 | 0 |
| Murg | 3.7 | 0 | 0 | 9 | 87 | 1 | 12 |
| Ova da Cluozza | 3.4 | 34 | 34 | 34 | 8 | 92 | 0 |
| Riale di Calneggia | 3.3 | 39 | 44 | 44 | 96 | 4 | 0 |
| Rietholzbach | 3.7 | 0 | 0 | 0 | 100 | 0 | 0 |
| Schaechen | 3.5 | 73 | 67 | 23 | 76 | 24 | 0 |
| Sense | 3.6 | 39 | 24 | 47 | 85 | 10 | 5 |
| Sitter | 3.2 | 71 | 61 | 36 | 48 | 52 | 0 |
| Vogelbach | 3.2 | 100 | 51 | 0 | 100 | 0 | 0 |

**An alternative interpolation method for precipitation isotopes (method 2)**

Precipitation $\delta^{18}$O measurements from 19 long-term monitoring stations in Switzerland (13 stations from NAQUA-ISOT, the Swiss network for Observations of Isotopes in the Water Cycle) and Germany (6 stations from GNIP, the Global Network of Isotopes in Precipitation) were decomposed into sine

5   functions and time series of residuals from the sine functions.

[Figure]

**Figure S 1: Locations of the 19 long-term monitoring stations for precipitation isotopes in Germany and Switzerland used for method 2, as well as the locations of the 22 study catchments in Switzerland (see Fig. 1 and Sect. 3 in the main text for a detailed description of the study catchments).**

10   The precipitation $\delta^{18}$O measurements $c(t)$ were fitted to sine curves through least squares regression:

$$c(t) = A\sin(2\pi f t - \varphi) + k \qquad\qquad\qquad (S1)$$

In Eq. (S1), $A$ is the amplitude (‰), $\varphi$ is the phase of the seasonal cycle (rad, with $2\pi$ rad equalling 1 year), $t$ is the time (decimal years), $f$ is the frequency (1 year$^{-1}$) and $k$ (‰) is a constant describing the vertical offset of the isotope signal. The mean RMSE for the sine fits across all measurement stations

15   was 2.1 ‰ $\delta^{18}$O.

Each of the three parameters describing the best-fit sine functions ($A$, $\varphi$, and $k$) of the 19 long-term monitoring stations were interpolated for all of Switzerland using multiple linear regression models based on latitudes, longitudes, and elevations:

$$A = 0.0002 \cdot \text{elevation} + 0.22 \cdot \text{longitude} - 0.88 \cdot \text{latitude} + 3.97 \quad , \qquad (S2)$$

20   $$\varphi = -3.47 \cdot 10^{-5} \cdot \text{elevation} + 0.007 \cdot \text{longitude} + 0.049 \cdot \text{latitude} - 1.82 \quad , \qquad (S3)$$

$$k = -0.0025 \cdot \text{elevation} - 0.38 \cdot \text{longitude} + 0.50 \cdot \text{latitude} - 10.4 \quad . \qquad (S4)$$

The explanatory variables in Eqs. (S2) - (S4) have been centered around their means, so that the intercepts describe the average latitudes, longitudes and elevations of the 19 stations, rather than an extrapolation to the arbitrary values latitude=0, longitude=0, and elevation=0.

The performance of the multiple-regression models that describe the spatial variations of the best-fit
5  sine functions was quantified by RMSE, $R^2$ and the $p$-values of the individual coefficients (Table S2):

**Table S2: RMSE, $R^2$ and the $p$-values of the individual coefficients of the multiple-regression models.**

|  | RMSE | $R^2$ | Elevation ($p$ value) | Longitude ($p$ value) | Latitude ($p$ value) | Intercept ($p$ value) |
|---|---|---|---|---|---|---|
| Amplitude $A$ | 0.70‰ | 0.56 | 0.62 | 0.16 | 0.004 | $1.6 \cdot 10^{-13}$ |
| Phase of the seasonal cycle $\varphi$ | 0.09rad | 0.29 | 0.51 | 0.72 | 0.15 | $5.8 \cdot 10^{-22}$ |
| Constant $k$ | 0.66‰ | 0.87 | 0.00001 | 0.01 | 0.06 | $3.25 \cdot 10^{-20}$ |

It should be noted that the three station properties were not strongly correlated with one another (i.e., $R=0.23$ and $p=0.35$ for elevation versus longitude; $R=-0.42$ and $p=0.07$ for elevation versus latitude;
10  $R=0.30$ and $p=0.21$ longitude versus latitude). The linear regression models were used to model sine parameters ($A$, $\varphi$, and $k$) for every 200m pixel in the 22 Swiss study catchments.

In a second step, the time series of residuals from the sine functions were geostatistically interpolated for every month of the time period 2010-2015 and every 200m pixel in the 22 Swiss study catchments. The spatial interpolation was carried out through ordinary kriging, applying an exponential variogram
15  model. Monthly maps of residuals from the sine functions were then used to adjust the base sinusoidal pattern for each 200m pixel in the 22 Swiss study catchments.

To quantify the prediction error of this interpolation method, it was run iteratively to simulate the monthly precipitation isotopic composition for each of the 19 long-term monitoring stations. For each of the 19 iterations, the precipitation isotope time series was predicted for one station by using only the
20  remaining 18 stations for calibration (i.e., a leave-one-out process). This two-step approach resulted in a 1.3 ‰ $\delta^{18}O$ mean absolute deviation between observations and model outputs (Figure S2).

[Figure]

**Figure S2: Modelled monthly isotope ($\delta^{18}$O) time series predicted for the 13 Swiss long-term monitoring stations (Figure S 1). The precipitation isotope time series were predicted for one station at a time by using only the remaining 18 stations (i.e., the other 12 Swiss stations and 6 German stations) for calibration (i.e., a leave-one-out process). Dots indicate the monthly observations, while lines indicate the modelled time series.**

Similar to interpolation method 1 (Seeger and Weiler, 2014), monthly isotope values obtained with method 2 were volume-weighted for each pixel based on the monthly elevation-dependent precipitation volumes obtained from the PREVAH model (Viviroli et al., 2009). Next, the monthly precipitation isotope values were aggregated across all 200m pixels in each catchment for a volume-weighted, catchment-averaged precipitation isotope time series. Snow accumulation and melt were not distinguished from liquid precipitation; that is to say, precipitation was treated as a direct input to the catchment at time of falling and snowpack storage was considered to be part of catchment storage (see Sect. 4.2 in main text).

The mass-weighted, catchment-averaged precipitation isotope time series were used for obtaining the parameter $A_P$ (Eqs. (1), (3), and (5) in the main text). For the 22 study catchments, the approach presented above resulted in different $A_P$ values than those obtained by method 1 (Seeger and Weiler, 2014); method 1 predicted higher $A_P$ values for higher elevation sites (Fig. 3 in the main text). In applying the alternative method described here, we find that elevation is a weak predictor of seasonal cycle amplitudes $A$ (Table S2). In contrast to method 1, we find that $A$ was primarily controlled by latitude and longitude, resulting in the largest $A_P$ values for catchments in south-eastern Switzerland (Dischmabach and Ova da Cluozza). However, spatial variations in $\delta^{18}O$ in precipitation are not simply a product of elevation (as in method 1) or of elevation, latitude, and longitude (method 2), because both methods presented here used kriging to incorporate other possible isotope effects.

**Table S 3: Long-term monitoring stations with their latitudes, longitudes and elevations used for the interpolation method presented here.**

| Long-term monitoring station | Latitude | Longitude | Elevation (m a.s.l.) |
|---|---|---|---|
| Sevelen (CH) | 47.12 | 9.49 | 457 |
| Grimsel (CH) | 46.57 | 8.33 | 1950 |
| Guttannen (CH) | 46.66 | 8.29 | 1055 |
| Meiringen (CH) | 46.73 | 8.18 | 632 |
| Belp (CH) | 46.90 | 7.51 | 515 |
| La Brevine (CH) | 46.98 | 6.61 | 1042 |
| Buchs-Suhr (CH) | 47.37 | 8.08 | 397 |
| Sion (CH) | 46.22 | 7.34 | 482 |
| Nyon (CH) | 46.38 | 6.23 | 436 |
| Locarno (CH) | 46.17 | 8.79 | 379 |
| Pontresina (CH) | 46.49 | 9.90 | 1742 |
| Basel (CH) | 47.54 | 7.58 | 319 |
| St.Gallen (CH) | 47.43 | 9.42 | 805 |
| Konstanz (GER) | 47.68 | 9.19 | 443 |
| Weil am Rhein (GER) | 47.60 | 7.59 | 249 |
| Karlsruhe (GER) | 49.04 | 8.37 | 112 |
| Hohenpeissenberg (GER) | 47.80 | 11.01 | 977 |
| Stuttgart (GER) | 48.83 | 9.20 | 314 |
| Garmisch-Partenkirchen (GER) | 47.48 | 11.06 | 719 |

---

## Author Response (AR2)

**Response to the second interactive comment of Reviewer #1 on**

"Sensitivity of young water fractions to hydro-climatic forcing and landscape properties across 22 Swiss catchments" by Jana von Freyberg et al.

Submitted on 14 Jun 2018

*The authors addressed all my comments and either included them in the revised manuscript or explained why they did not. I think the new paper structure significantly improved the readability of the manuscript. Thank you for this highly interesting and relevant contribution. I have just got a few minor comments that you might want to consider before resubmission:*

We thank Reviewer#1 for these comments, which we have addressed in detail below.

*Comments of the reviewer are shown in italics.* Responses from the authors are presented in regular font below each comment. Citations from the manuscript are in Times New Roman, changes of the manuscript text are underlined.

*Main text (final version without tracked changes)*

*• Please update all instances of "Allen et al.,submitted manuscript" as this paper has been published by now.*
We have done that.

*• Page 4, lines 13-14: "Young water fractions have so far only been used in a global analysis of 254 watersheds...". This is true on the global scale, but there are a few local studies using Fyw, including Song et al., Hydrological Processes, 31(4), 935-947, 2017, doi:10.1002/hyp.11077, and Stockinger et al., Journal of Hydrology 541, 952–964, 2016, doi:10.1016/j.jhydrol.2016.08.007.*
*Please mention these as well.*
We have included these references.

*• Page 10, line 3: suggest changing to "we apply the methods 1 and 2 (section 3.4) for interpolating...." to clarify you are referring to the two methods you introduced before.*
We have changed that.

*• Page 16, line 21: suggest adding "The linear slope was determined from linear regression of eq. (10) and the uncertainty in this slope was estimated..." (if I got it right?) to specify how you calculated the discharge sensitivity "directly from the tracer time series".*
The linear slope was not determined through linear regression. The linear slope of the Q-$F_{yw}$-relationship is equivalent to $m_s/A_p$ in Eq. (10), as it is stated on Page 16 Line 21. Since we estimated the slope from Eq. (10), we propagated the errors of $F_{yw}(Q)$, $n_s$ and $A_p$ in order to obtain the uncertainty in the slope $m_s/A_p$.

*• Page 17, lines 9-10: good to see you were able to include the short comment. I'd suggest changing to "These values are similar to the "precipitation sensitivities of Fyw" found by Wilusz et al. (2017) for two neighbouring catchments in Plynlimon, Wales." as "those" now reads as if Wilusz et al. (2017) calculated discharge sensitivities as well.*
We have reformulated this sentence to "A similar analysis was carried out by Wilusz et al. (2017) for two neighbouring catchments in Plynlimon, Wales. For those two sites,…"

*• Page 17, lines 5 and 13: d-1 instead of day-1 for consistency reasons*

Thank you for catching this. We have updated the units.

*Figures*
*• Figure 3 - caption: "those of Seeger and Weiler (2014) and those presented in the Supplement (methods 1 and 2, respectively)". This sounds as if you have used more than two methods. But If I understood correctly, you are comparing method 1 (according to Seeger and Weiler, 2014) and method 2.*
We have changed that.

*• Figure 6: I cannot see any changes to this figure. I'd still recommend increasing the contrast between light blue and grey dots (maybe also with other symbols) as in comparison to the other (very nice) figures, the symbols in this figure are not easily discernible.*
We have increased the contrast.

*• Figure 9: I'd suggest having the same colour coding from low to high values (instead of a reverse coding) for panels a and b to allow for better comparison between Fyw and its discharge sensitivity.*
We have changed the color scheme.

*Supplement*
*• Page 6, lines 11-12: (Seeger and Weiler, 2014), which predicted higher Ap values..."*
Thank you for catching this. We have corrected that error.

[revised manuscript text omitted]

*Correspondence to*: Jana von Freyberg (jana.vonfreyberg@usys.ethz.ch)

- Table S1 with additional data about the phases of the seasonal precipitation regimes, as well as hydrologic soil properties and hydrogeological characteristics of the individual study sites
- Detailed description of an alternative interpolation method for precipitation isotopes (method 2)
- *R* script for performing iteratively reweighted least squares (IRLS) regression with optional point weights, including a demo data set ("*IRLS_hess-2017-720.R*")

**Table S1: Elevation ranges, as well as hydrologic soil properties and hydrogeological characteristics of the 22 Swiss study catchments.**

| Catchment name | Phase of seasonality of monthly precipitation $\varphi_{precip}$ (months) | Fraction of shallow soils (%) | Fraction of low - medium water storage capacity soils (%) | Fraction of high - very high permeability soils (%) | Fraction of aquifers with low productivity (%) | Fraction of aquifers with intermediate productivity (%) | Fraction of aquifers with high productivity (%) |
|---|---|---|---|---|---|---|---|
| Aabach | 3.8 | 0 | 0 | 23 | 87 | 0 | 13 |
| Aach | 3.9 | 0 | 0 | 0 | 86 | 0 | 14 |
| Allenbach | 3.2 | 78 | 57 | 30 | 89 | 11 | 0 |
| Alp | 3.2 | 68 | 48 | 1 | 81 | 6 | 12 |
| Biber | 3.3 | 30 | 30 | 0 | 94 | 0 | 6 |
| Dischmabach | 3.5 | 59 | 59 | 59 | 92 | 9 | 0 |
| Emme | 3.3 | 78 | 49 | 21 | 88 | 11 | 0 |
| Ergolz | 4.0 | 42 | 41 | 28 | 42 | 54 | 5 |
| Erlenbach | 3.2 | 100 | 4 | 0 | 82 | 18 | 0 |
| Guerbe | 3.6 | 48 | 35 | 45 | 70 | 17 | 13 |
| Ilfis | 3.3 | 32 | 28 | 42 | 92 | 1 | 7 |
| Langeten | 3.4 | 0 | 0 | 37 | 77 | 13 | 10 |
| Lümpenenbach | 3.1 | 100 | 4 | 0 | 100 | 0 | 0 |
| Mentue | 4.7 | 0 | 0 | 76 | 99 | 0 | 0 |
| Murg | 3.7 | 0 | 0 | 9 | 87 | 1 | 12 |
| Ova da Cluozza | 3.4 | 34 | 34 | 34 | 8 | 92 | 0 |
| Riale di Calneggia | 3.3 | 39 | 44 | 44 | 96 | 4 | 0 |
| Rietholzbach | 3.7 | 0 | 0 | 0 | 100 | 0 | 0 |
| Schaechen | 3.5 | 73 | 67 | 23 | 76 | 24 | 0 |
| Sense | 3.6 | 39 | 24 | 47 | 85 | 10 | 5 |
| Sitter | 3.2 | 71 | 61 | 36 | 48 | 52 | 0 |
| Vogelbach | 3.2 | 100 | 51 | 0 | 100 | 0 | 0 |

**An alternative interpolation method for precipitation isotopes (method 2)**

Method 2 for the spatial interpolation of precipitation isotopes is based on the approach developed by Allen et al. (2018), and is briefly described here. Precipitation $\delta^{18}$O measurements from 19 long-term monitoring stations in Switzerland (13 stations from NAQUA-ISOT, the Swiss network for

5 Observations of Isotopes in the Water Cycle) and Germany (6 stations from GNIP, the Global Network of Isotopes in Precipitation) were decomposed into sine functions and time series of residuals from the sine functions.

[Figure]

Figure S 1: Locations of the 19 long-term monitoring stations for precipitation isotopes in Germany and Switzerland used for
10 method 2, as well as the locations of the 22 study catchments in Switzerland (see Fig. 1 and Sect. 3 in the main text for a detailed description of the study catchments).

The precipitation $\delta^{18}$O measurements $c(t)$ were fitted to sine curves through least squares regression:

$$c(t) = A\sin(2\pi f t - \varphi) + k \tag{S1}$$

In Eq. (S1), $A$ is the amplitude (‰), $\varphi$ is the phase of the seasonal cycle (rad, with $2\pi$ rad equalling

15 1 year), $t$ is the time (decimal years), $f$ is the frequency (1 year$^{-1}$) and $k$ (‰) is a constant describing the vertical offset of the isotope signal. The mean RMSE for the sine fits across all measurement stations was 2.1 ‰ $\delta^{18}$O.

Each of the three parameters describing the best-fit sine functions ($A$, $\varphi$, and $k$) of the 19 long-term monitoring stations were interpolated for all of Switzerland using multiple linear regression models

20 based on latitudes, longitudes, and elevations:

$$A = 0.0002 \cdot \text{elevation} + 0.22 \cdot \text{longitude} - 0.88 \cdot \text{latitude} + 3.97 \quad , \tag{S2}$$

$$\varphi = -3.47 \cdot 10^{-5} \cdot \text{elevation} + 0.007 \cdot \text{longitude} + 0.049 \cdot \text{latitude} - 1.82 \quad , \tag{S3}$$

$$k = -0.0025 \cdot \text{elevation} - 0.38 \cdot \text{longitude} + 0.50 \cdot \text{latitude} - 10.4 \quad . \tag{S4}$$

The explanatory variables in Eqs. (S2) - (S4) have been centered around their means, so that the intercepts describe the average latitudes, longitudes and elevations of the 19 stations, rather than an

5    extrapolation to the arbitrary values latitude=0, longitude=0, and elevation=0.

The performance of the multiple-regression models that describe the spatial variations of the best-fit sine functions was quantified by RMSE, $R^2$ and the $p$-values of the individual coefficients (Table S2):

**Table S2: RMSE, $R^2$ and the $p$-values of the individual coefficients of the multiple-regression models.**

|  | RMSE | $R^2$ | Elevation ($p$ value) | Longitude ($p$ value) | Latitude ($p$ value) | Intercept ($p$ value) |
|---|---|---|---|---|---|---|
| Amplitude $A$ | 0.70‰ | 0.56 | 0.62 | 0.16 | 0.004 | $1.6 \cdot 10^{-13}$ |
| Phase of the seasonal cycle $\varphi$ | 0.09rad | 0.29 | 0.51 | 0.72 | 0.15 | $5.8 \cdot 10^{-22}$ |
| Constant $k$ | 0.66‰ | 0.87 | 0.00001 | 0.01 | 0.06 | $3.25 \cdot 10^{-20}$ |

10    It should be noted that the three station properties were not strongly correlated with one another (i.e., $R$=0.23 and $p$=0.35 for elevation versus longitude; $R$=-0.42 and $p$=0.07 for elevation versus latitude; $R$=0.30 and $p$=0.21 longitude versus latitude). The linear regression models were used to model sine parameters ($A$, $\varphi$, and $k$) for every 200m pixel in the 22 Swiss study catchments.

In a second step, the time series of residuals from the sine functions were geostatistically interpolated

15    for every month of the time period 2010-2015 and every 200m pixel in the 22 Swiss study catchments. The spatial interpolation was carried out through ordinary kriging, applying an exponential variogram model. Monthly maps of residuals from the sine functions were then used to adjust the base sinusoidal pattern for each 200m pixel in the 22 Swiss study catchments.

To quantify the prediction error of this interpolation method, it was run iteratively to simulate the

20    monthly precipitation isotopic composition for each of the 19 long-term monitoring stations. For each of the 19 iterations, the precipitation isotope time series was predicted for one station by using only the remaining 18 stations for calibration (i.e., a leave-one-out process). This two-step approach resulted in a 1.3 ‰ $\delta^{18}$O mean absolute deviation between observations and model outputs (Figure S2).

[Figure]

**Figure S2: Modelled monthly isotope (δ¹⁸O) time series predicted for the 13 Swiss long-term monitoring stations (Figure S 1). The precipitation isotope time series were predicted for one station at a time by using only the remaining 18 stations (i.e., the other 12 Swiss stations and 6 German stations) for calibration (i.e., a leave-one-out process). Dots indicate the monthly observations, while lines indicate the modelled time series.**

Similar to interpolation method 1 (Seeger and Weiler, 2014), monthly isotope values obtained with method 2 were volume-weighted for each pixel based on the monthly elevation-dependent precipitation volumes obtained from the PREVAH model (Viviroli et al., 2009). Next, the monthly precipitation isotope values were aggregated across all 200m pixels in each catchment for a volume-weighted, catchment-averaged precipitation isotope time series. Snow accumulation and melt were not distinguished from liquid precipitation; that is to say, precipitation was treated as a direct input to the catchment at time of falling and snowpack storage was considered to be part of catchment storage (see Sect. 4.2 in main text).

The mass-weighted, catchment-averaged precipitation isotope time series were used for obtaining the parameter $A_P$ (Eqs. (1), (3), and (5) in the main text). For the 22 study catchments, the approach presented above resulted in different $A_P$ values than those obtained by method 1 (Seeger and Weiler, 2014), which predicted higher $A_P$ values for higher elevation sites (Fig. 3 in the main text). In applying the alternative method described here, we find that elevation is a weak predictor of seasonal cycle amplitudes $A$ (Table S2). In contrast to method 1, we find that $A$ was primarily controlled by latitude and longitude, resulting in the largest $A_P$ values for catchments in south-eastern Switzerland (Dischmabach and Ova da Cluozza). However, spatial variations in $\delta^{18}O$ in precipitation are not simply a product of elevation (as in method 1) or of elevation, latitude, and longitude (method 2), because both methods presented here used kriging to incorporate other possible isotope effects.

**Table S 3: Long-term monitoring stations with their latitudes, longitudes and elevations used for the interpolation method presented here.**

| Long-term monitoring station | Latitude | Longitude | Elevation (m a.s.l.) |
|---|---|---|---|
| Sevelen (CH) | 47.12 | 9.49 | 457 |
| Grimsel (CH) | 46.57 | 8.33 | 1950 |
| Guttannen (CH) | 46.66 | 8.29 | 1055 |
| Meiringen (CH) | 46.73 | 8.18 | 632 |
| Belp (CH) | 46.90 | 7.51 | 515 |
| La Brevine (CH) | 46.98 | 6.61 | 1042 |
| Buchs-Suhr (CH) | 47.37 | 8.08 | 397 |
| Sion (CH) | 46.22 | 7.34 | 482 |
| Nyon (CH) | 46.38 | 6.23 | 436 |
| Locarno (CH) | 46.17 | 8.79 | 379 |
| Pontresina (CH) | 46.49 | 9.90 | 1742 |
| Basel (CH) | 47.54 | 7.58 | 319 |
| St.Gallen (CH) | 47.43 | 9.42 | 805 |
| Konstanz (GER) | 47.68 | 9.19 | 443 |
| Weil am Rhein (GER) | 47.60 | 7.59 | 249 |
| Karlsruhe (GER) | 49.04 | 8.37 | 112 |
| Hohenpeissenberg (GER) | 47.80 | 11.01 | 977 |
| Stuttgart (GER) | 48.83 | 9.20 | 314 |
| Garmisch-Partenkirchen (GER) | 47.48 | 11.06 | 719 |